# Proteomic and functional comparison between human induced and embryonic stem cells

Alejandro J Brenes[1,2,3]*[†], Eva Griesser[1‡], Linda V Sinclair[2], Lindsay Davidson[3], Alan R Prescott[4], Francois Singh[5§], Elizabeth KJ Hogg[5], Carmen Espejo-Serrano[5], Hao Jiang[1], Harunori Yoshikawa[1#], Melpomeni Platani[1], Jason R Swedlow[1], Greg M Findlay[5], Doreen A Cantrell[2], Angus I Lamond[1]*

[1]Molecular, Cell and Developmental Biology, School of Life Sciences, University of Dundee, Dundee, United Kingdom; [2]Cell Signalling & Immunology, School of Life Sciences, University of Dundee, Dundee, United Kingdom; [3]Human Pluripotent Stem Cell Facility, School of Life Sciences, University of Dundee, Dow St, Dundee DD1 5EH, Dundee, United Kingdom; [4]Dundee Imaging Facility, School of Life Sciences, University of Dundee, Dundee, United Kingdom; [5]MRC Protein Phosphorylation and Ubiquitylation Unit, School of Life Sciences, University of Dundee, Dundee, United Kingdom

*For correspondence:
abrenes@ed.ac.uk (AJB);
a.i.lamond@dundee.ac.uk (AIL)

Present address: [†]Centre for Inflammation Research, Institute for Regeneration and Repair University of Edinburgh, Edinburgh, United Kingdom; [‡]Drug Discovery Sciences, Boehringer Ingelheim Pharma GmbH & Co, Biberach an der Riss, Germany; [§]Department of Physiology, Faculty of Medicine, Biomedical Center, University of Iceland, Reykjavík, Iceland; [#]Division of Cell Signalling, Fujii Memorial Institute of Medical Sciences, Institute of Advanced Medical Sciences, Tokushima University, Tokushima, Japan

## eLife Assessment

This study reports differences in proteomic profiles of embryonic versus induced pluripotent stem cells. This **important** finding cautions against the interchangeable use of both types of cells in biomedical research, although the mechanisms responsible for these differences remains unknown. The proteomic evidence is **convincing**, even though there is limited validation with other methods.

**Abstract** Human induced pluripotent stem cells (hiPSCs) have great potential to be used as alternatives to embryonic stem cells (hESCs) in regenerative medicine and disease modelling. In this study, we characterise the proteomes of multiple hiPSC and hESC lines derived from independent donors and find that while they express a near-identical set of proteins, they show consistent quantitative differences in the abundance of a subset of proteins. hiPSCs have increased total protein content, while maintaining a comparable cell cycle profile to hESCs, with increased abundance of cytoplasmic and mitochondrial proteins required to sustain high growth rates, including nutrient transporters and metabolic proteins. Prominent changes detected in proteins involved in mitochondrial metabolism correlated with enhanced mitochondrial potential, shown using high-resolution respirometry. hiPSCs also produced higher levels of secreted proteins, including growth factors and proteins involved in the inhibition of the immune system. The data indicate that reprogramming of fibroblasts to hiPSCs produces important differences in cytoplasmic and mitochondrial proteins compared to hESCs, with consequences affecting growth and metabolism. This study improves our understanding of the molecular differences between hiPSCs and hESCs, with implications for potential risks and benefits for their use in future disease modelling and therapeutic applications.

## Introduction

Human embryonic stem cells (hESCs) are derived from the inner cell mass of a pre-implantation embryo (*Smith, 2001*). They show prolonged undifferentiated potential, as well as the ability to differentiate into the three main embryonic germ layers (*Thomson et al., 1998*), making them excellent models for studying disease mechanisms, development, and differentiation. However, their use remains restricted by regulations, based in part upon ethical considerations (*Volarevic et al., 2018*).

Over a decade ago, methods allowing the induction of pluripotent stem cells from fibroblast cultures, in both human and mice, were developed (*Takahashi et al., 2007*; *Takahashi and Yamanaka, 2006*). These reports showed that by exogenously expressing a small set of key transcription factors (Oct4, Sox2, c-Myc, and Klf4), a somatic cell could be reprogrammed back into a pluripotent state, characterised by their capacity for self-renewal and ability to differentiate into the three main germ layers. These human induced pluripotent stem cells (hiPSCs) show many key features of their physiological hESCs counterparts, while avoiding many of the ethical issues regarding the use of stem cells derived from embryos.

Since the discovery of reprogramming methods, hiPSC lines have attracted great interest, particularly for their potential use as alternatives to hESCs in regenerative medicine (*Kimbrel and Lanza, 2015*) and disease modelling, including studies on monogenic disorders (*Ebert et al., 2009*; *Lee et al., 2009*) and some late onset diseases (*Liu et al., 2012*). However, to understand the value of using hiPSCs in regenerative therapy, drug development and/or studies of disease mechanisms, it is important to establish how similar hiPSCs are to hESCs at the molecular and functional levels. To address this, multiple studies have compared hiPSCs and hESCs, using a variety of assays, including methylation analysis (*Mallon et al., 2014*), transcriptomics (*Mallon et al., 2013*; *Guenther et al., 2010*), and even quantitative proteomics (*Munoz et al., 2011*; *Phanstiel et al., 2011*). It should be noted, however, that many of these earlier studies were performed at a time when reprogramming protocols were less robust (*Vitale et al., 2012*) and when the depth of proteome coverage and quantitative information that could be obtained was lower than today.

In this study, we have addressed the similarity of hiPSCs to hESCs by performing a detailed proteomic analysis, comparing a set of four hiPSC lines derived from human primary skin fibroblasts (*Kilpinen et al., 2017*) of independent, healthy donors, with four independent hESC lines. The data highlight that while both types of stem cell lines have very similar global protein abundance profiles, they also show some specific and significant quantitative differences in protein expression. In particular, the reprogrammed iPSC lines consistently display higher total protein levels, predominantly affecting cytoplasmic proteins required to sustain higher growth, along with mitochondrial changes, and an excess of secreted proteins, with impact upon cell phenotypes.

## Results

### hESCs and hiPSCs display quantitative differences in protein abundances

For this study, we compared multiple hESC and hiPSC lines, all derived from different donors and cultured using identical growth conditions. First, the expression levels of the main pluripotency markers were verified in each of the lines, with no differences detected between the respective hESC and hiPSC types (*Figure 1a*). From these data, representative sets of four hiPSCs and four hESCs lines were selected for detailed proteomic analysis using mass spectrometry (MS). The proteomes were characterised using tandem mass tags (TMT) (*Thompson et al., 2003*), within a single 10-plex (TMT channel allocation) and using MS3-based synchronous precursor selection (SPS) (*McAlister et al., 2014*). To further optimise quantification accuracy, each sample was allocated to a specific isobaric tag to minimise cross-population reporter ion interference (*Figure 1b*), as previously described (*Brenes et al., 2019*). In total 8491 protein groups (henceforth referred to as 'proteins') were detected at 1% false discovery rate (FDR), with >99% overlap between the proteins detected from both the hESC and hiPSC lines (*Figure 1c*). However, it is important to highlight that TMT is not the best method to use when looking for proteins that are specific to one condition or population, as protein detected in one channel are frequently seen across all other channels as well (*Brenes et al., 2019*).

To provide a quantitative comparison of the respective proteomes, we focussed on analysing the 7878 proteins that were detected with at least two unique and razor peptides (see Materials

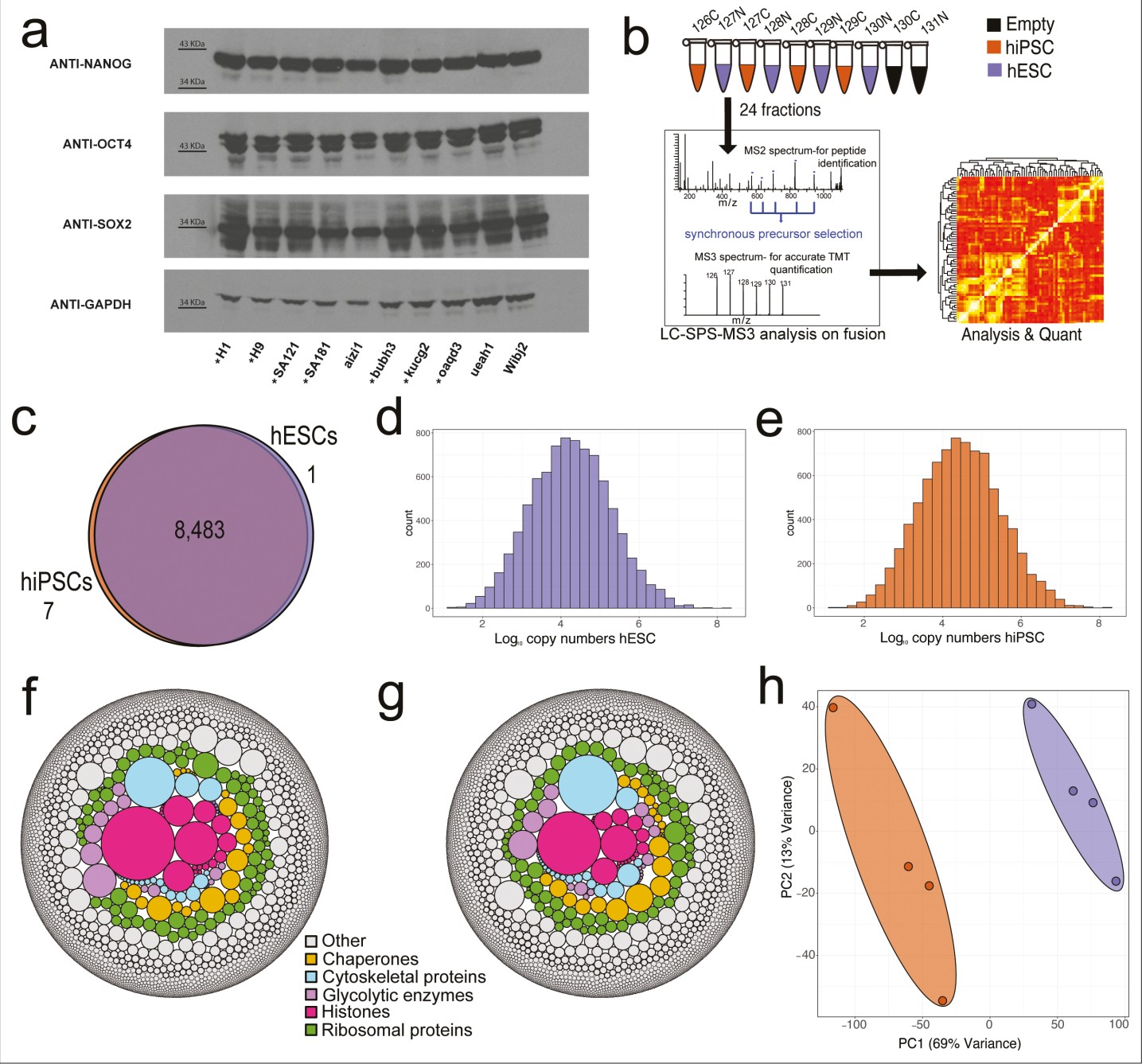

**Figure 1.** Proteomic overview. (**a**) Western blots showing the expression of the pluripotency factors NANOG, OCT4, and SOX2 across all human embryonic stem cell (hESC) and human induced pluripotent stem cell (hiSPC) lines. The eight lines showed with * were used within the proteomic analysis. (**b**) Diagram showing the SPS-MS3 tandem mass tag (TMT) proteomic workflow used for the experiment. (**c**) Venn diagram showing the overlap of proteins identified within the hiPSC and hESC populations. (**d**) Average copy number histogram for the hESCs (N=4). (**e**) Average copy number histogram for the hiPSCs (N=4). (**f**) Bubble plot showing proteins coloured by specific categories where the size is represented by the average hESC estimated protein copy numbers. (**g**) Bubble plot showing proteins coloured by specific categories where the size is represented by the average hiPSC estimated protein copy numbers. (**h**) Principal component analysis (PCA) plot based on the log$_{10}$ copy numbers for all eight stem cell lines. hESCs are shown in purple and hiPSCs in orange.

The online version of this article includes the following source data and figure supplement(s) for figure 1:

**Source data 1.** PDF containing the western blots for *Figure 1a*.

**Source data 2.** Original files for the western blots displayed in *Figure 1a*.

**Figure supplement 1.** Box plot showing the sum of the normalised intensity for all histones across human embryonic stem cells (hESCs) and human induced pluripotent stem cells (hiPSCs).

and methods). After confirming that there were no differences in the abundance levels of histones between the two cell types (*Figure 1—figure supplement 1*), protein copy numbers were estimated via the 'proteomic ruler' (*Wiśniewski et al., 2014*) (see Materials and methods). The copy number data (*Supplementary file 1*) highlighted that both the hESC (*Figure 1d*) and hiPSC (*Figure 1e*) proteomes display a similar dynamic range, with estimated protein copy numbers extending from a median of less than 100 copies, to over 100 million copies per cell. Furthermore, the composition of the respective proteomes is highly similar. Both cell types display high expression levels of ribosomal proteins, protein chaperones, and glycolytic enzymes (*Figure 1f and g*), consistent with rapid proliferation and dependence on glycolysis for energy generation (*Folmes et al., 2011*). It is only when the quantitative data are examined in more detail that differences between the cell types become apparent (*Figure 1h*). A principal component analysis (PCA), based on the protein copy numbers, revealed a clear separation between the two stem cell populations within the main component of variation, which accounted for 69% of variance. The PCA suggested that the independent hiPSC lines were clearly different to the hESC lines, and vice versa.

## Standard normalisation methods mask changes in total protein content in hiPSCs compared to hESCs

A previous proteomic study reported that there were virtually no protein level differences between hESCs and hiPSCs (*Munoz et al., 2011*). However, in that study the intensity data were median normalised. We therefore decided to compare two different normalisation methods: i.e., the previously used median normalisation method and the 'proteomic ruler' (*Wiśniewski et al., 2014*). The median normalisation produces concentration-like results and is frequently used to normalise proteomic data. With this approach, our data also show no major differences in protein abundances between the hESC and hiPSC lines (*Figure 2a*; *Supplementary file 2*), i.e., ~94% of all proteins displayed no significant changes in abundance (FC>1.5-fold; q-value<0.001), similar to the previously reported conclusion (*Munoz et al., 2011*). However, median (or total intensity) normalisation methods lack the capacity to detect changes in absolute abundance, cell size, or protein content. By artificially forcing all medians to be almost identical, such changes are invisible.

This is not the case for the results produced with the 'proteomic ruler' (*Wiśniewski et al., 2014*). The copy number-based analysis enables an approximation to absolute protein abundance and can reveal changes in cell mass, as we previously reported (*Howden et al., 2019*; *Marchingo et al., 2020*). Using the proteomic ruler method highlighted systematic differences between hESCs and hiPSCs (*Figure 2b*; *Supplementary file 3*), with 56% (4426/7878) of all proteins detected significantly increased in hiPSCs (FC>1.5-fold; q-value<0.001) and with particular enrichment in translation-related processes (*Figure 2—figure supplement 1*). In contrast, only 40 proteins (0.5%) showed significantly lower expression levels in hiPSCs. With thousands of proteins displaying higher abundance, we hypothesised that hiPSCs have higher total protein content, compared to hESCs. Using the protein copy numbers to estimate the total protein content showed that hiPSCs had >50% higher protein content compared to hESCs (*Figure 2c*). To validate this observation, an independent assay (EZQ assay; see Materials and methods) was used to measure the total protein yield from similar numbers of freshly grown hiPSCs and hESCs. From these experiments, the calculated protein amount per million cells was >70% higher (*Figure 2d*; p-value=0.0018) in hiPSCs, relative to hESCs. We conclude that hiPSCs have a higher total protein content.

Changes in protein content could potentially be linked to differences in the cell cycle profile. Hence, to test this, we used fluorescence-activated cell sorting (FACS) to study the cell cycle distribution of hESCs and hiPSCs. The FACS data showed that hiPSCs have significantly higher forward scatter (*Figure 2e*), correlated to increased cell size, as well as significantly higher side scatter (*Figure 2f*), correlating to increased cell granularity. However, the FACS analysis revealed no significant differences between hiPSCs and hESCs in the percentage of cells at each of the cell cycle stages (*Figure 2e*). We conclude that hiPSCs have significantly higher total protein content, with increased size and granularity, but that these differences with hESCs are independent of changes in cell cycle distribution.

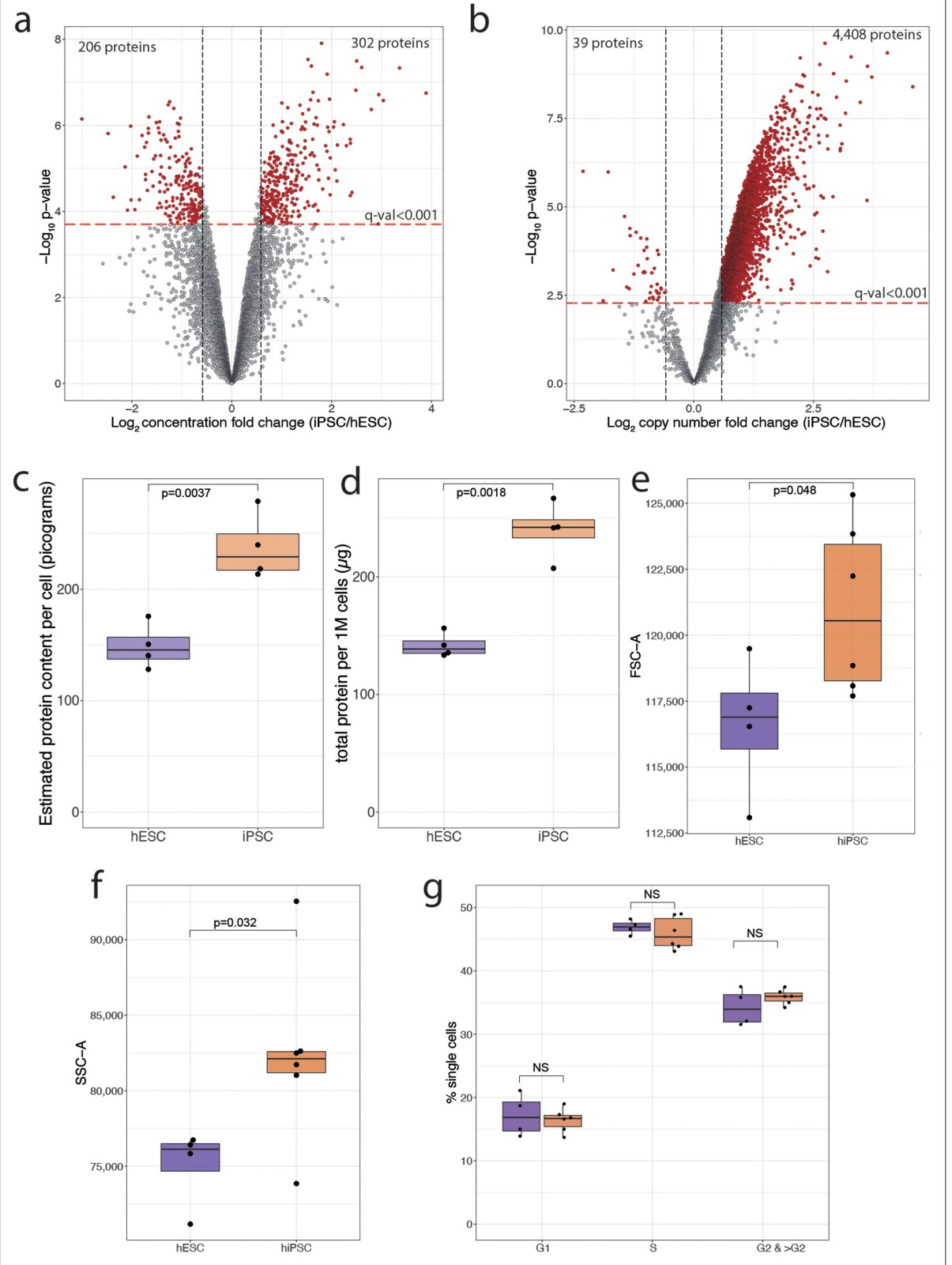

**Figure 2.** Normalisation and protein content. (**a**) Concentration-based volcano plot showing the -$\log_{10}$ p-value and the $\log_2$ fold change comparing human induced pluripotent stem cells (hiPSCs) (N=4) to human embryonic stem cells (hESCs) (N=4). Elements shaded in red are considered significantly changed. All dots above the red line have a q-value lower than 0.001. (**b**) Copy number-based volcano plot showing the -$\log_{10}$ p-value and the $\log_2$ fold change comparing hESCs (N=4) to hiSPC (N=4). Elements shaded in red are considered significantly changed. All dots above the red line have

*Figure 2 continued on next page*

*Figure 2 continued*

a q-value lower than 0.001. (**c**) Boxplot showing the mass spectrometry (MS)-based estimated protein content for hESCs (N=4) and hiPSC(N=4). (**d**) Boxplot showing the protein amount per million cells derived from the EZQ Protein Quantification Kit for all hESCs (N=4) and hiPSC (N=4). (**e**) Boxplot showing the median forward scatter of hESCs (N=4) and hiPSCs (N=6). (**f**) Boxplot showing the median side scatter of hESCs (N=4) and hiPSCs (N=6). (**g**) Boxplot showing the median percentage of cells across cell cycle stages for hESCs (N=4) and hiPSCs (N=6). For all boxplots, the bottom and top hinges represent the first and third quartiles. The top whisker extends from the hinge to the largest value no further than 1.5 × IQR from the hinge; the bottom whisker extends from the hinge to the smallest value at most 1.5 × IQR of the hinge.

The online version of this article includes the following figure supplement(s) for figure 2:

**Figure supplement 1.** Barplot showing the results of a Gene Ontology Biological Process overrepresentation analysis for proteins that are significantly increased in human induced pluripotent stem cells (hiPSCs) compared to human embryonic stem cells (hESCs) in the copy number analysis.

## hiPSCs have elevated nutrient transporters, metabolic proteins, and protein and lipid synthesis machinery

To maintain a higher protein content than hESCs with a comparable cell cycle profile, hiPSCs would require higher protein synthesis capacity, which in turn requires increased access to nutrients and energy. Energy metabolism in primed pluripotent stem cells is largely dependent on glycolysis (*Turner et al., 2014*), which is sensitive to glucose uptake and lactate shuttling. Therefore, we compared the expression of the respective glucose and lactate transporters between hiPSCs and hESCs. The data showed both main glucose transporters, GLUT1 (SLC2A1) and GLUT3 (SLC2A3), had higher abundance in hiPSCs, as did the lactate transporters SLC16A1 and SLC16A3 (*Figure 3a*). Other rate limiting enzymes, including Hexokinase 1 (HK1) and 2 (HK2), were also significantly increased within hiPSCs (*Figure 3b*), suggesting increased glycolytic potential.

Nutrient uptake is mostly handled by the SLC (solute carrier), group of membrane transporters. Analysis of the 15 SLC transporters that are most upregulated in hiPSCs, compared to hESCs, showed that they mostly belonged to two categories, i.e., amino acid and mitochondrial transporters (*Figure 3c*). Amino acids are vital to sustain high rates of protein synthesis (*Marchingo and Cantrell, 2022*) and the data showed that 11/12 amino acid transporters were significantly increased in hiPSCs, compared to hESCs, including the hyper abundant protein SLC3A2, which is present at >4 million copies per cell (*Figure 3d*). The highest fold increases (>4-fold) were seen for SLC38A1 and SLC38A2, both of which are major glutamine transporters (*Bhutia and Ganapathy, 2016*; *Bröer et al., 2016*).

We next examined whether the increased abundance of the glutamine transporters had phenotypic impact, i.e., whether it correlated with increased glutamine uptake within hiPSCs. To test this hypothesis, we measured the uptake of radiolabelled glutamine in both hiPSCs and hESCs (see Materials and methods). The data showed that hiPSCs had a median of >90% higher uptake of glutamine, compared to hESCs (*Figure 3e*). Glutamine has been reported to be the most consumed amino acid in hESCs (*Marsboom et al., 2016*) and its catabolism to be one of the vital metabolic pathways that can provide ATP and more importantly biosynthetic precursors, required to sustain growth (*Tohyama et al., 2016*). Hence, we also explored the abundance of enzymes involved in glutaminolysis and found that vital proteins, including GLS, GLUD1, GPT2, and GOT2, were also significantly higher in hiPSCs (*Figure 3f*).

Having established that hiPSCs have increased expression of nutrient transporters and higher expression of enzymes in key metabolic pathways, compared with hESCs, we next looked at the machinery required for protein synthesis. The levels of many of the proteins involved in ribosome subunit biogenesis, including ribosomal proteins, were higher in hiPSCs (*Figure 3g*). The increased expression of translation machinery components, nutrient transporters, and many metabolic enzymes is consistent with the increased total protein content seen within hiPSCs.

The data also highlighted increased fatty acid and lipid droplet (LD) synthesis potential in hiPSCs, with increased abundance of the protein SREBP1 (SREBF1; *Figure 3h*), a master regulator of lipid synthesis (*Eberlé et al., 2004*), as well as FASN (*Figure 3i*) and SCD (*Figure 3j*). Similarly, a crucial regulator for LD assembly, PLIN3 (*Nose et al., 2013*, *Figure 3k*), displayed >2-fold increased abundance in hiPSCs. To examine the potential phenotypic impact of this increased abundance of proteins involved in LD synthesis, we performed transmission electron microscopy (TEM) analyses to compare hiPS and hES cells. This showed that LDs were clearly visible in hiPSCs (*Figure 3l*), but not visible in hESCs (*Figure 3m*). We conclude that the hiPSCs have elevated levels of LDs, resulting from the increased expression of proteins involved in lipid synthesis and LD assembly.

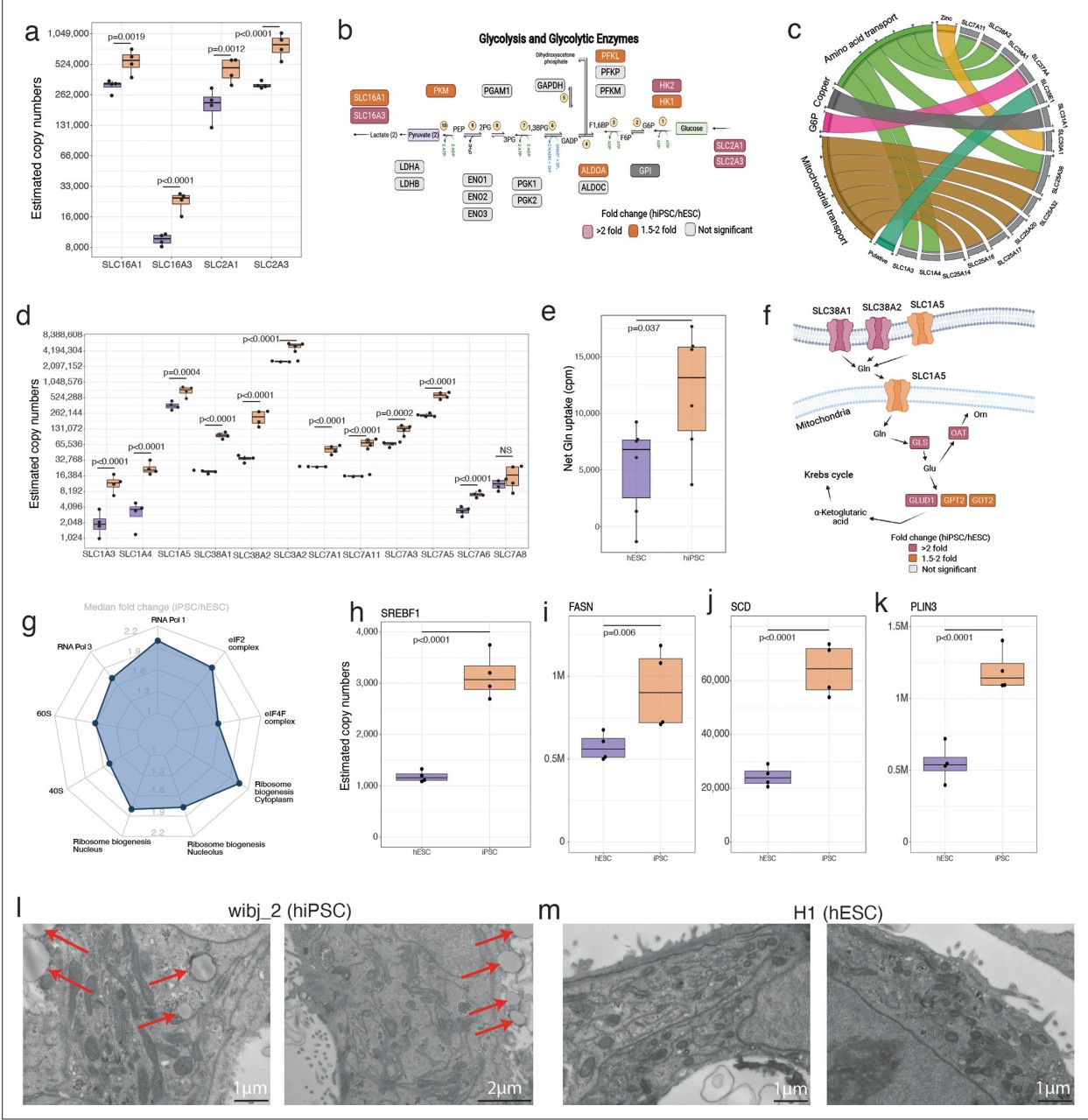

**Figure 3.** Fuelling growth. (**a**) Boxplots showing the estimated copy numbers for the lactate (SLC16A1 and SLC16A3) and glucose transporters (SLC2A1 and SLC2A3) across human embryonic stem cells (hESCs) (N=4) and human induced pluripotent stem cell (hiPSC) (N=4). (**b**) Schematic showing the glycolytic proteins and their fold change in hESC vs hiPSCs. This panel was created using BioRender.com. (**c**) Chord diagram showing the 15 most upregulated solute carrier proteins along with their classification based on transport activities/localisation. (**d**) Boxplots showing the estimated copy numbers of the main amino acid transporters in hESCs (N=4) and hiPSC (N=4). (**e**) Boxplot showing the net glutamine uptake (see Materials and methods) in hESCs (N=6) and hiPSC (N=6). (**f**) Schematic showing the glutaminolysis proteins and their fold change in hESCs (N=4) and hiPSC (N=4). This panel was created using BioRender.com. (**g**) Radar plot showing the median fold change (hiPSC/ESC) for protein categories which are related to the pre-ribosomes. Boxplots showing the estimated copy numbers for (**h**) SREBF1, (**i**) FASN, (**j**) SCD, (**k**) PLIN3 in hESCs (N=4) and hiPSC (N=4). (**l**) Transmission electron microscopy images for wibj_2 (hiPSC). Lipid droplets are marked with red arrows. (**m**) Transmission electron microscopy images for H1 (hESC). For all boxplots, the bottom and top hinges represent the first and third quartiles. The top whisker extends from the hinge to the largest value no further than 1.5 × IQR from the hinge; the bottom whisker extends from the hinge to the smallest value at most 1.5 × IQR of the hinge.

## hiPSCs show altered mitochondrial metabolism compared to hESCs

Our data also highlighted important changes in mitochondrial proteins, including increases in the levels of metabolic proteins that are encoded within the mitochondrial genome (*Taanman, 1999*, *Figure 4a*). The latter proteins, which are translated by specialised mitochondrial ribosomes (mitoribosomes), are embedded in the mitochondrial membrane. The protein components of mitoribosomes also showed increased expression in hiPSCs (*Figure 4b*), along with virtually all proteins involved in the translation initiation, elongation, and termination of mitochondrial genome-encoded proteins (*Figure 4c*).

The analysis of transporter proteins revealed a cluster of 22/27 mitochondrial transporters that were significantly increased in hiPSCs, including the hyper abundant (>10 million copies per cell), ATP/ADP transporter (*Figure 4d*). A subset of 14 transporters displayed >2-fold increased abundance, including the acylcarnitine transporter SLC25A20 (*Figure 4e*), which is part of the carnitine shuttle in the β-oxidation pathway. Another component of the carnitine shuttle, CPT1A, displayed over fourfold higher abundance in hiPSCs (*Figure 4f*), suggesting an important role. The data showed it was not just fatty acid oxidation, but also synthesis, that was affected, with proteins acting in the mitochondrial fatty acid synthesis (mFAS) pathway also increased in abundance. MCAT (*Figure 4g*), MECR (*Figure 4h*), and OXSM (*Figure 4i*), all displayed ~2-fold higher abundance in hiPSCs compared to hESCs. These results have a metabolic relevance as mFAS has been reported to control the activity of the electron transport chain (ETC) (*Nowinski et al., 2020*), which was also increased in hiPSCs, with subunits of all five ETC complexes increased in abundance in hiPSCs and with complex II and complex III showing the most prominent effects (*Figure 4j*). Complex II is also part of the tricarboxylic acid (TCA) cycle, which displayed increased abundance of the majority of proteins involved in the pathway (*Figure 4k*).

As the proteomic data showed clear differences between hiPSCs and hESCs in the levels of mitometabolism proteins, we performed experiments to explore whether this was reflected in phenotypic differences between hiPSCs and hESCs. This was tested using high-resolution respirometry (see Materials and methods). The data showed that hiPSCs had a higher P/E control ratio to hESCs, which denotes an increased capacity of the phosphorylation system to produce ATP (*Figure 4l*). We conclude that hiPSCs have elevated levels of mitometabolism proteins relative to hESCs, resulting in higher respiratory activity.

## hiPSCs upregulate secreted proteins affecting their microenvironment

Among the most upregulated proteins in hiPSCs were a subset of secreted proteins. Secreted proteins are of great importance because changes in their absolute abundance can affect the extracellular environment. These secreted proteins mostly represented four categories: structural extracellular matrix (ECM) proteins, growth factors, protease inhibitors, and proteases (*Figure 5a*). The ECM both provides physical support for cells and actively participates in cell signalling by providing domains for growth factors (*Mouw et al., 2014*). The ECM can also be reshaped in tumours, thereby promoting cancer cell growth and migration (*Rømer et al., 2021*). The current data show that both laminins and collagens were all increased in abundance in hiPSCs (*Figure 5b*). Collagens are reported to alter the stiffness of the ECM and their synthesis is iron intensive. Interestingly, the data also show that proteins involved in importing and storing iron were increased in abundance in hiPSCs (*Figure 5c–f*).

The data also showed that 13 growth factors were increased in abundance in hiPSCs, compared to hESCs. A subset of these, i.e., FGF1, FGF2, and NODAL, are reported to have direct relevance to the maintenance of pluripotency and can modulate important processes in PSCs (*Lanner and Rossant, 2010*; *Xu et al., 2008*; *Weinberger et al., 2016*, *Figure 5g*). Other growth factors that are upregulated in hPSCs are linked to disease mechanisms and cancer. This includes VGF (*Figure 5h*), which is linked to promoting growth and survival in glioblastoma (*Wang et al., 2018*) and MDK (*Figure 5i*), which is highly expressed in malignant tumours (*Filippou et al., 2020*) and has been shown to play a role in chemoresistance (*Lu et al., 2018*).

## hiPSCs display increased abundance of immunosuppressive proteins

NODAL wasn't the only growth factor in the TGFB family that was increased in hiPSCs, with TGFB1 displaying an ~5-fold increase in abundance in hiPSCs compared to hESCs (*Figure 5j*). Besides its role as a growth factor, TGFB1 has been shown to have important roles in the regulation of the immune response, promoting the generation of regulatory T cells, while inhibiting the generation and function

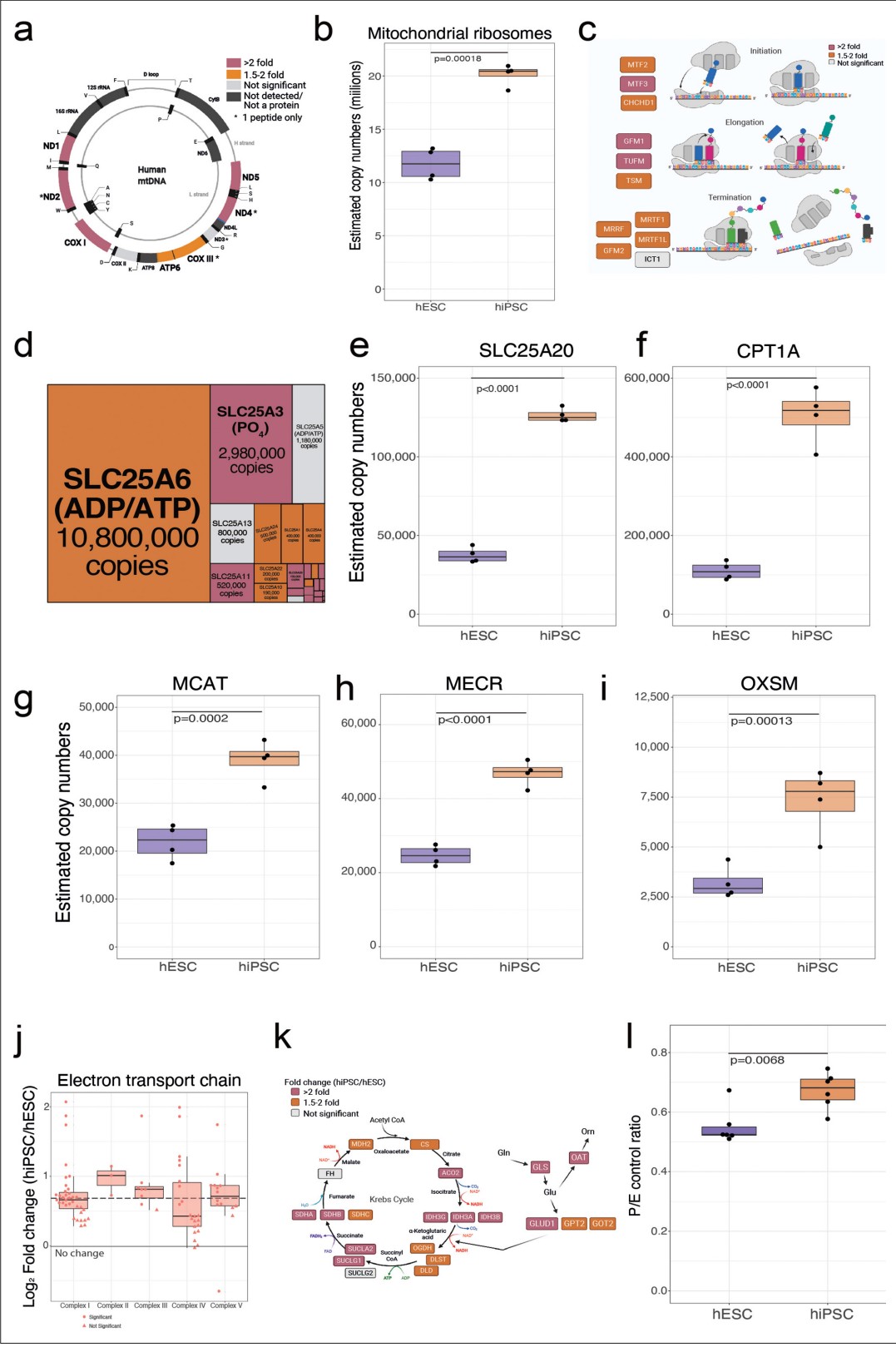

**Figure 4.** Mitochondrial differences. (**a**) Schematic showing the mitochondrial genome encoded proteins and their fold change in human embryonic stem cells (hESCs) and human induced pluripotent stem cells (hiPSCs). This panel was created using BioRender.com. (**b**) Boxplot showing the estimated copy numbers of all mitochondrial ribosomal proteins hESCs (N=4) and hiPSC (N=4). (**c**) Schematic showing proteins involved in mitochondrial translation

*Figure 4 continued on next page*

*Figure 4 continued*

and their fold change (hiPSCs/hESCs). This panel was created using BioRender.com. (**d**) Treeplot showing all mitochondrial transporters, size is proportional to the median estimated copy numbers in hiPSCs (N=4). Boxplot showing the estimated copy numbers for (**e**) SLC25A20. (**f**) CPT1A. (**g**) MCAT, (**h**) MECR, (**i**) OSXM, in hESCs (N=4) and hiPSC (N=4). (**j**) Boxplot showing the $\log_2$ fold change (hiPSC/hESCs) for all subunits of the different complexes of the electron transport chain. The median fold change across all detected proteins is shown as a dotted line. (**k**) Schematic showing the fold change of critic acid cycle and glutaminolysis proteins in hiPSCs (N=4) vs hESCs (N=4). This panel was created using BioRender.com. (**l**) Boxplot showing the P (oxphos capacity)/E (electron transfer capacity) control ratio in wibj_2 (hiPSC; N=6) vs H1 (hESC; N=6). For all boxplots, the bottom and top hinges represent the first and third quartiles. The top whisker extends from the hinge to the largest value no further than $1.5 \times$ IQR from the hinge; the bottom whisker extends from the hinge to the smallest value at most $1.5 \times$ IQR of the hinge.

of effector T cells. As immunogenicity of PSCs is a topic of relevance to clinical adaptations, we looked for differences in modulators of the immune response.

Arginine availability is vital to effector T cells and other leukocytes, where depletion mediated by arginase has been shown to be linked to T cell inhibition (*Vonwirth et al., 2020*). Our data show that hiPSCs have ~2.5-fold higher abundance of ARG1 (*Figure 5k*). Furthermore, hiPSCs also display increased expression of the immune checkpoint protein, CD276 (*Figure 5l*), which has been reported to be a potent inhibitor of survival and function of T cells (*Wang et al., 2021*; *Yue et al., 2021*).

hiPSCs also displayed increased abundance of inhibitory ligands that suppress the immune function of other leukocytes. The data show hiPSCs have increased abundance of the non-classical HLA-E (*Figure 5m*), which has been shown to interact with the NK cell receptor, NKG2A, to mediate immune evasion in ageing cells (*Pereira et al., 2019*). They also displayed increased abundance of CD200 (*Figure 5n*), a ligand for CD200R, which can inhibit the immune response from macrophages, basophils, NK cells, and T cells, as well as CD47 (*Figure 5o*), a ligand of SIRPA that helps cells to escape macrophage phagocytosis. These data indicate that hiPSCs have increased abundance of known immunosuppressive proteins, compared to hESCs.

## hiPSCs display reduced abundance of H1 histones

A striking feature of this proteomic study is how few proteins (<1%; 40/7878) showed significantly decreased abundance in hiPSCs, compared to hESCs. A high proportion of these decreased abundance proteins affect nuclear processes. Thus, an overrepresentation analysis showed that proteins whose abundance was decreased in hiPSCs were enriched in GO terms related to DNA recombination, nucleosome positioning, and chromatin silencing (*Figure 6a*). Notably, this included four H1 histone variants, which are reported to influence nucleosomal repeat length (*Woodcock et al., 2006*) and stabilise chromatin structures (*Robinson and Rhodes, 2006*). Our data show that the most abundant H1 variant in hESCs, HIST1H1E, is decreased in abundance in hiPSCs by ~3.5-fold (*Figure 6b*), while HIST1H1C (*Figure 6c*), HIST1H1D (*Figure 6d*), and H1FX (*Figure 6e*) are all decreased by >1.7-fold.

As histone variants have very similar protein sequences, where peptides detected by MS can potentially match to multiple H1 histone variants, a peptide level analysis was necessary to deconvolute the signal (*Figure 6—figure supplement 1*). The Andromeda search engine (*Cox et al., 2011*) assigns peptide intensities to a protein following a razor peptide approach, where the intensity of a peptide is assigned to only one protein, regardless of whether it is unique or shared by two or more proteins. This makes the analysis of specific variants challenging at the protein level. Hence, we focussed on a peptide-specific approach with the MS data and found that the intensity of the peptides that were shared between these 3 H1 histone variants displayed a consistent reduction in abundance in hiPSCs (*Figure 6—figure supplement 1*).

The systematic reduction in abundance in hiPSCs seen with H1 histone variants was not seen for members of the other histone families. Evaluating either the concentration (*Figure 1—figure supplement 1*), or copy numbers (*Figure 6f*), across all histones, showed no significant differences in expression between hiPSCs and hESCs. Furthermore, for core histones, including H3 and H4, there were no significant abundance differences seen within either the proteomics data (*Figure 6g and h*), or in additional western blot analyses that were performed to validate these conclusions (*Figure 6i and j*). However, we did detect differences between hiPSCs and hESCs in the expression of histone H2 variants, with H2AFV (*Figure 6k*), H2AFY (*Figure 6l*), and H2AFY2 (*Figure 6m*), all increased in abundance

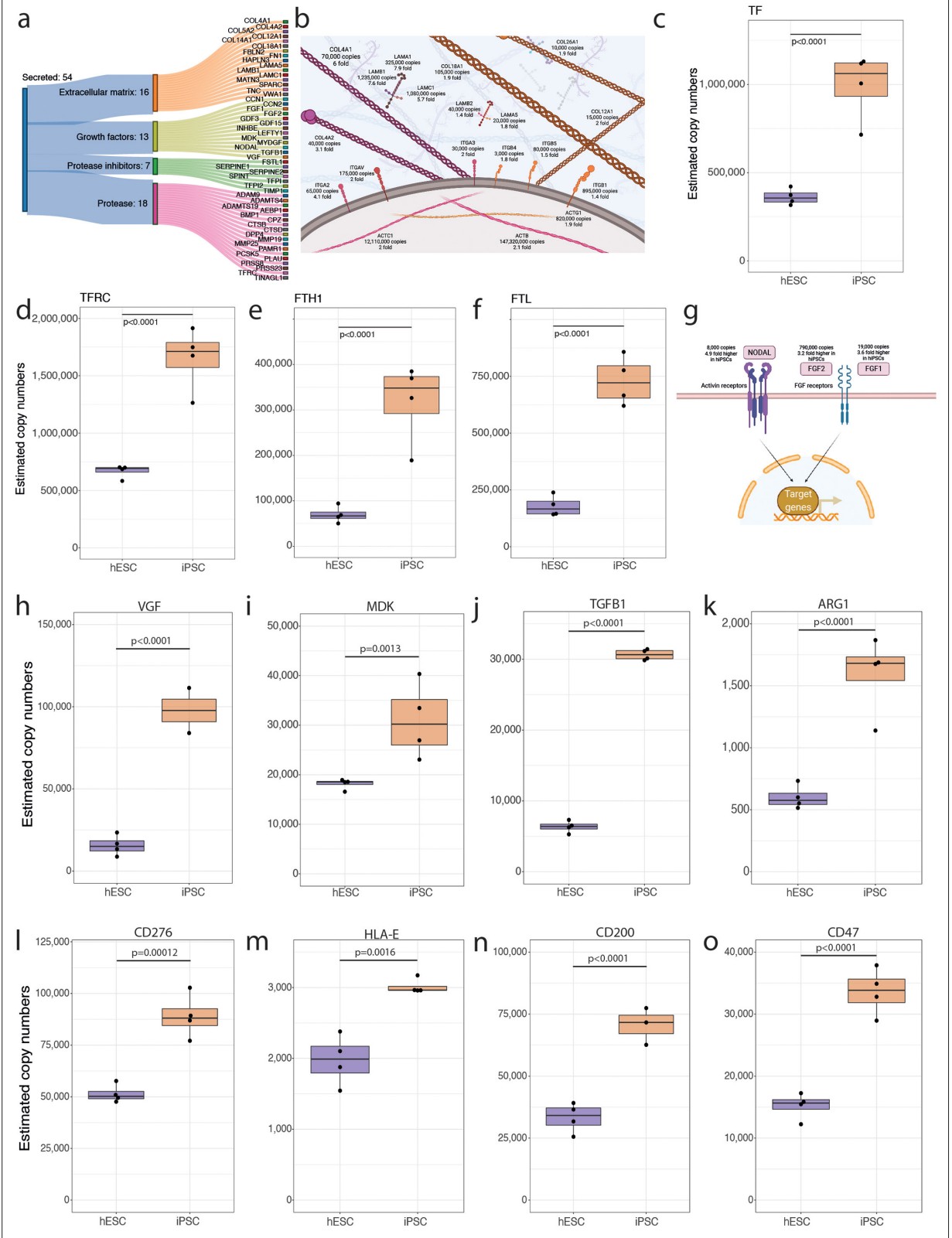

**Figure 5.** Secreted proteins. (**a**) Sankey diagram showing the secreted proteins that belong to the extracellular matrix (ECM), growth factor, protease inhibitor, or protease categories and are significantly increased in abundance in human induced pluripotent stem cells (hiPSCs). (**b**) Schematic showing ECM proteins that are significantly increased in abundance in hiPSCs. Boxplot showing the estimated copy numbers for (**c**) TF, (**d**) TFRC, (**e**) FTH1, and (**f**) FTL in human embryonic stem cells (hESCs) (N=4) and hiPSC (N=4). (**g**) Schematic showing the changes in abundance in primed pluripotency growth

*Figure 5 continued on next page*

Figure 5 continued

factors. (**h**) Boxplot showing the estimated protein copy numbers for VGF. (**i**) Boxplot showing the estimated protein copy numbers for MDK. Boxplot showing the estimated protein copy numbers for (**j**) TGFB1, (**k**) ARG1, (**l**) CD276, (**m**) HLA-E, (**n**) CD200, and (**o**) CD47. All boxplots show the data for hESCs and hiPSCs. For all boxplots, the bottom and top hinges represent the first and third quartiles. The top whisker extends from the hinge to the largest value no further than 1.5 × IQR from the hinge; the bottom whisker extends from the hinge to the smallest value at most 1.5 × IQR of the hinge.

in hiPSCs. As histone H2 variants also have high sequence similarity and shared peptides, we also performed a peptide level analysis, which validated that both shared and unique peptides displayed the same pattern, i.e., showing increased abundance in hiPSCs (*Figure 6—figure supplement 2*). Thus, we conclude that there are opposing effects for histone H1 and histone H2 variants, with the former decreased and the latter increased in abundance in hiPSCs.

## Discussion

Induced pluripotent stem cells can provide valuable models for clinical research and future therapies, which makes it vital to understand both their similarities and any specific differences, with embryo-derived human stem cells. This study provides a detailed comparison of the proteomes of multiple hiPSC and hESC lines derived from different donors. The major conclusion is that while hiPSC and hESC lines express a near-identical set of proteins, with similar abundance ranks, they also display important quantitative differences. In particular, our data indicate that hiPSCs reprogrammed from skin fibroblasts display considerable differences in their cytoplasmic and mitochondrial proteomes, compared to hESCs, while the nuclear proteome was very similar between the two cell types. Furthermore, additional microscopy analyses and functional assays showed that the systematic differences in the proteomes of the respective hiPSCs and hESCs had a measurable impact on cell phenotypes, most notably affecting mitochondria, metabolic activity, and nutrient transport. It would be of interest in future to analyse whether hiPSCs reprogrammed from different cell types, such as peripheral leuko-cytes, also share these differences in protein expression with hESCs, or if these specific changes are characteristic of fibroblast-derived iPS cells.

Using estimated protein copy numbers, our data show that only <1% of proteins were signifi-cantly decreased in abundance in hiPSCs, compared to hESCs, including multiple H1 histones. In contrast, ~56% of all proteins quantified were significantly increased in hiPSCs (fold change>1.5 and q-value<0.001), with most of these increases affecting cytoplasmic and mitochondrial proteins and activities. The MS data show that total protein levels are higher overall in hiPSCs, as compared with hESCs, a result that was independently validated using an EZQ protein assay. This difference in total protein content was shown by FACS analysis not to result from systematic differences between hiPSCs and hESCs in cell cycle progression. Instead, the increased protein levels in hiPSCs correlated with increased levels of the protein translational machinery, along with increased metabolic and mitochon-drial activity and higher levels of nutrient transport.

These results highlight an important technical point relating to data normalisation and its effect on the interpretation of such data. By using a standard median normalisation (concentration-based approach), instead of the proteomic ruler (*Wiśniewski et al., 2014*), the difference in total protein content between the cell types, involving the increased abundance of thousands of proteins, is not apparent. Hence, two cell types with, for example, a fourfold higher total protein content, could none-theless appear to show virtually no significant differences, as long as the protein ranks and relative concentrations remained similar. This would result in an erroneous conclusion that there is little to no change in protein expression, while the orthogonal data suggest otherwise.

Having established that hiPSCs displayed higher total protein content than corresponding hESC lines, we sought to understand how this could be maintained. To maximise protein synthesis, nutrient availability and energy production are key factors (*Marchingo and Cantrell, 2022*). The proteomic data show that vital nutrient transporters, known to be important for growth and protein production (*Bröer, 2020*), were significantly increased in hiPSCs, compared to hESCs. In particular, the three glutamine transporters, SLC1A5, SLC38A1, and SLC38A2, were all significantly increased in abun-dance, with additional functional assays showing that this correlated with higher levels of glutamine uptake measured in hiPSCs. Glutamine has been previously shown to fuel growth and proliferation in

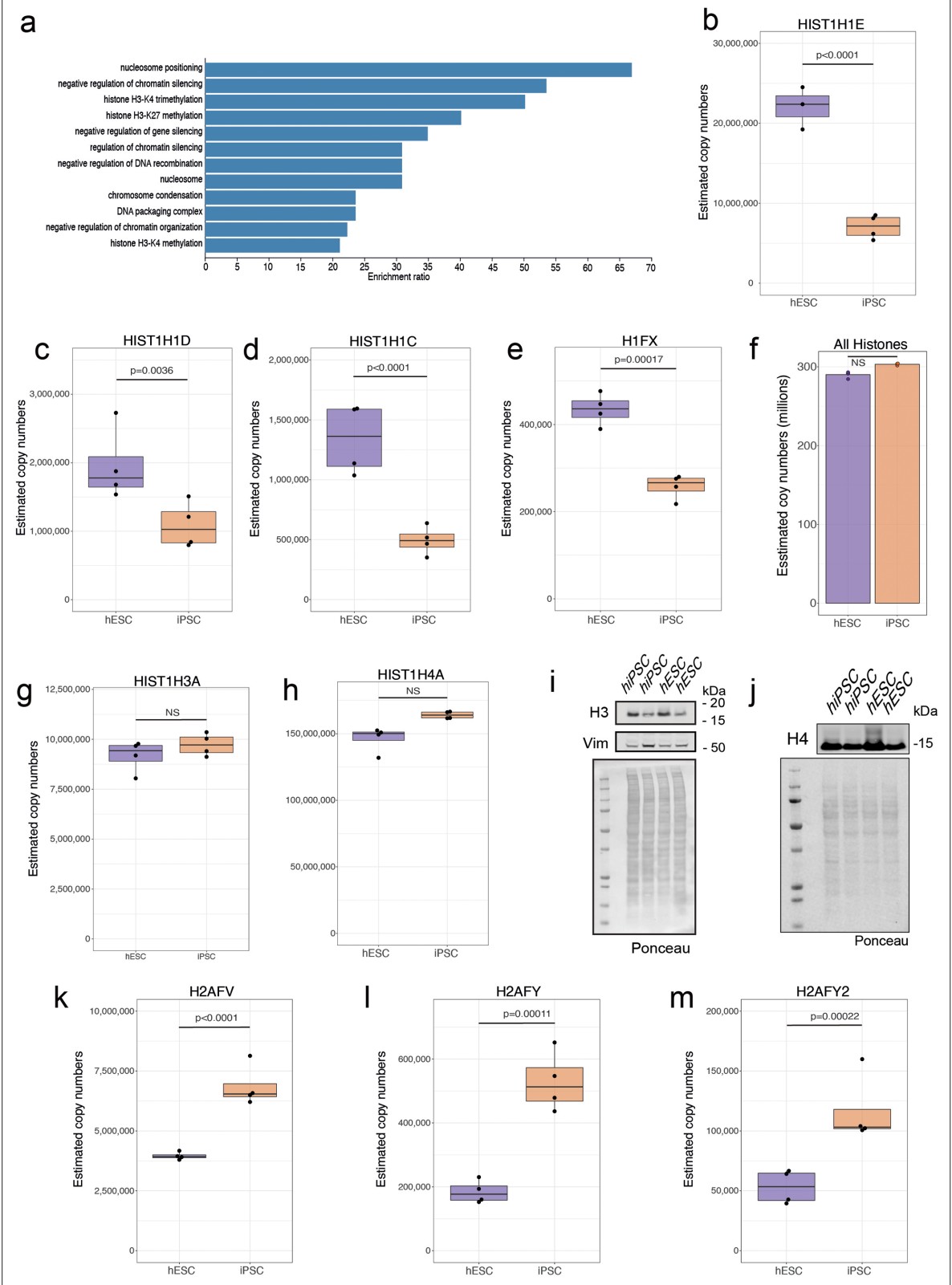

**Figure 6.** Changes within histones. (**a**) Barplot showing the GO term enrichment results for proteins significantly decreased in abundance (see Materials and methods) in human induced pluripotent stem cells (hiPSCs). Boxplots showing estimated copy numbers for (**b**) HIST1H1E, (**c**) HIST1H1D, (**d**) HIST1H1C, and (**e**) H1FX in human embryonic stem cells (hESCs) (N=4) and hiPSC (N=4). (**f**) Barplot showing the median estimated copy numbers for all histones in hESCs and hiPSCs. Boxplots showing estimated copy numbers for (**g**) HIST1H3A and (**h**) HIST1H4A in hESCs and hiPSCs. Western blot

*Figure 6 continued on next page*

*Figure 6 continued*

showing the abundance of (**i**) H3 and (**j**) H4 histones in hESCs (N=4) and hiPSC (N=4). Boxplots showing estimated copy numbers for (**k**) H2AFV, (**l**) H2AFY, and (**m**) H2AFY2 in hESCs (N=4) and hiPSC (N=4). For all boxplots, the bottom and top hinges represent the first and third quartiles. The top whisker extends from the hinge to the largest value no further than 1.5 × IQR from the hinge; the bottom whisker extends from the hinge to the smallest value at most 1.5 × IQR of the hinge.

The online version of this article includes the following source data and figure supplement(s) for figure 6:

**Source data 1.** PDF containing the western blots for *Figure 6i and j*.

**Source data 2.** Original files for the western blots displayed in *Figure 6i and j*.

**Figure supplement 1.** Schematic showing the unique and shared peptides detected for H1 histones as well as boxplots showing the $\log_2$ histone normalised intensity for both human embryonic stem cells (hESCs) and human induced pluripotent stem cells (hiPSCs).

**Figure supplement 2.** Schematic showing the unique and shared peptides detected for H2AFV as well as boxplots showing the $\log_2$ histone normalised intensity for both human embryonic stem cells (hESCs) and human induced pluripotent stem cells (hiPSCs).

rapidly dividing cells, including cancer cells (*Bhutia and Ganapathy, 2016*) and could be sustaining higher rates in hiPSCs.

Nutrients provide the fuel, but it is the metabolic proteins that are the engines that convert them to energy. Here, our data showed that proteins involved in both glycolysis and glutaminolysis were significantly increased in abundance in hiPSCs. When cells preferentially use the glycolytic pathway, e.g., stem cells and cancer cells, there is increased demand for biosynthetic precursors and NADPH (*Ju et al., 2020*). These precursors can be supplied via glutaminolysis (*Jin et al., 2016*; *Wise and Thompson, 2010*; *Reitzer et al., 1979*) linked to the TCA, both important mitochondrial processes and both with significantly increased protein levels in hiPSCs, along with other proteins involved in the ETC. One caveat is that an uneven distribution of biological sex between donors of the respective hESC and hiPSCs lines could mean there is a potential for sex-specific differences to explain at least some of these effects, however in previous a large-scale proteomic study of male and female hiPSCs no changes in proteins involved in oxidative phosphorylation were detected (*Brenes et al., 2021*). Differences in the mitochondria between hiPSCs and hESCs have been previously reported, but whether they originate from the reprogramming process, or are induced by the increased nutrient uptake, remains unknown and is a point of interest for future studies.

Secreted proteins, such as growth factors and ECM proteins, are a category of great interest, because their absolute abundance can affect the surrounding cellular microenvironment. hiPSCs were found here to show increased expression levels of growth factors that are linked to cancer and immunosuppression. For example, FGF2, an important growth factor for primed pluripotent stem cells, has been shown to promote ERK activation (*Lotz et al., 2013*), stimulating protein synthesis (*Ma et al., 2005*; *Pelletier et al., 2015*; *Galan et al., 2014*). Thus, the increased abundance of FGF2 could generate a feedforward loop, further driving/sustaining growth in hiPSCs, however that growth potential is also linked to breast (*Giulianelli et al., 2019*) and gastric cancers (*Li et al., 2020*), as well as gliomas (*Sooman et al., 2015*). Another important growth factor that is increased in abundance in hiPSCs is TGFB1, a known potent inhibitor of T cell responses (*Thomas and Massagué, 2005*; *Li and Flavell, 2008*). We note that the immunogenicity of pluripotent stem cells has important consequences for cell therapy applications. Our data suggest that hiPSCs might have a higher immune evasion potential, via multiple mechanisms. They display increased abundance of secreted T cell inhibitors, including TGFB1 and ARG1 (*Munder et al., 2006*), along with inhibitory ligands, such as CD276, CD200, and CD47. An increased inhibitory capacity, combined with the tumourigenic potential of hiPSCs (*Lee et al., 2013*), raises some concerns about the suitability of using reprogrammed hiPSCs for certain types of therapeutic applications. Based upon our current data, we recommend that potential phenotypic consequences resulting from the observed differences in iPS and ESC proteomes should be studied further to determine whether the immunosuppressive properties of these cells vary significantly.

In summary, our data show that hiPSCs and hESCs, despite their clear similarities, are not identical at both the protein and phenotypic levels. We show that hiPSCs reprogrammed from skin fibroblasts differ from hESCs, predominantly in their cytoplasmic and mitochondrial proteome, leading to measurable functional differences affecting their metabolic activity and growth potential. These data

can help to inform future strategies to mitigate for these differences as hiPSCs continue to be used in important clinical applications and as disease models.

## Materials and methods

### Key resources table

| Reagent type (species) or resource | Designation | Source or reference | Identifiers | Additional information |
|---|---|---|---|---|
| Cell line (*Homo sapiens*) | wibj_2 | HipSci Consortium (https://www.hipsci.org/) | RRID:CVCL_AE65 | |
| Cell line (*Homo sapiens*) | kucg_2 | HipSci Consortium (https://www.hipsci.org/) | RRID:CVCL_AE60 | |
| Cell line (*Homo sapiens*) | bubh_3 | HipSci Consortium (https://www.hipsci.org/) | RRID:CVCL_AE78 | |
| Cell line (*Homo sapiens*) | aqd_3 | HipSci Consortium (https://www.hipsci.org/) | RRID:CVCL_EE62 | |
| Cell line (*Homo sapiens*) | ueah_1 | HipSci Consortium (https://www.hipsci.org/) | RRID:CVCL_AG29 | |
| Cell line (*Homo sapiens*) | aizi_1 | HipSci Consortium (https://www.hipsci.org/) | RRID:CVCL_AG87 | |
| Cell line (*Homo sapiens*) | H1 (WA01) | WiCell Research Institute (https://www.wicell.org/) | RRID:CVCL_9771 | |
| Cell line (*Homo sapiens*) | H9 (WA09) | WiCell Research Institute (https://www.wicell.org/) | RRID:CVCL_9773 | |
| Cell line (*Homo sapiens*) | SA121 | Takara Bio Europe AB (Cellartis) | RRID:CVCL_B296 | |
| Cell line (*Homo sapiens*) | SA181 | Takara Bio Europe AB (Cellartis) | RRID:CVCL_B299 | |
| Software, algorithm | MaxQuant | https://www.maxquant.org/ | RRID:SCR_014485 | |
| Software, algorithm | LIMMA | http://bioinf.wehi.edu.au/limma/ | RRID:SCR_010943 | |
| Software, algorithm | qvalue | Bioconductor | RRID:SCR_001073 | |
| Software, algorithm | FlowJo | TreeStar | | |
| Antibody | Anti-Histone H3 (rabbit polyclonal) | Abcam | RRID:AB_302613 | (1:1000) |
| Antibody | Anti-Histone H4 (rabbit polyclonal) | Abcam | RRID:AB_296888 | (1:1000) |
| Antibody | Anti-Vimentin (rabbit monoclonal) | Cell Signaling Technology | RRID:AB_10695459 | (1:1000) |
| Antibody | Anti-GAPDH (mouse monoclonal) | Cell Signaling Technology | RRID:AB_2756824 | (1:10,000) |
| Antibody | Anti-OCT4A (rabbit monoclonal) | Cell Signaling Technology | RRID:AB_2167691 | (1:10,000) |
| Antibody | Anti-SOX2 (rabbit monoclonal) | Cell Signaling Technology | RRID:AB_2195767 | (1:10,000) |
| Antibody | Anti-NANOG (rabbit monoclonal) | Cell Signaling Technology | RRID:AB_10559205 | (1:10,000) |

### HipSci hiPSC line generation

As part of the HipSci project (*Kilpinen et al., 2017*) hiPSC lines were generated in the Sanger Centre. hiPSCs were generated from fibroblasts obtained by skin punch biopsies. The fibroblasts were reprogrammed as described previously (*Kilpinen et al., 2017*), in brief Sendai vectors expressing hOCT3/4, hSOX2, and hc-MYC were used.

### HipSci hiPSC line quality control

The hiPSC lines were passaged a mean of 16 times before being subjected to the first tier of molecular data for quality control, which included genotyping ('gtarray'), gene expression data ('gexarray'), and an assessment of the pluripotency and differentiation potential of each line ('Cellomics'). Pluripotency of the lines was additionally verified in silico, using the PluriTest assay (*Müller et al., 2011*). Subsequently one or two lines per donor were subjected to a set of molecular data QC assays. The criteria for line selection were: (i) level of pluripotency, as determined by the PluriTest assay, (ii) number

of copy number abnormalities, and (iii) ability to differentiate into each of the three germ layers. These included proteomics, DNA methylation ('mtarray'), RNA-sequencing, and high-content cellular imaging.

## Cell line authentification

STR profiling was conducted for H1, H9, SA121, and SA181, the profiles matched to available profiles on Cellosaurus. HipSci cell lines (aizi_1, bubh_3, kucg_2, oaqd_3, ueah_1 and wibj_2) were directly received from the HipSci consortium with the details provided above. The cell lines were routinely tested for mycoplasma contamination using MycoAlert Detection Kit (Lonza), aerobic bacteria and fungi were tested by inoculation of conditioned cell culture medium into tryptic soy broth (Millipore).

## hiPSC and hESC culture

hiPSCs generated by the HipSci consortium (*Kilpinen et al., 2017*) (aizi_1, bubh_3, kucg_2, oaqd_3, ueah_1 and wibj_2) and hESCs (SA121 and SA181, H1, H9) were both grown in identical conditions, maintained in TESR medium (*Ludwig et al., 2006*) supplemented with FGF2 (Peprotech, 30 ng/ml) and noggin (Peprotech, 10 ng/ml) on growth factor reduced Geltrex basement membrane extract (Life Technologies, 10 µg/cm$^2$) coated dishes at 37°C in a humidified atmosphere of 5% $CO_2$ in air.

Cells were routinely passaged twice a week as single cells using TrypLE select (Life Technologies) and replated in TESR medium that was further supplemented with the Rho kinase inhibitor Y27632 (Tocris, 10 µM) to enhance single-cell survival. Twenty-four hours after replating Y27632 was removed from the culture medium. For proteomic analyses cells were plated in 100 mm Geltrex-coated dishes at a density of $5\times10^4$ cells/cm$^2$ and allowed to grow for 3 days until confluent with daily medium changes.

## Immunoblotting

Equal volumes of hiPSC or hESCs protein lysates were boiled in LDS/RA buffer for 5 min at 95°C and loaded into 4–15% NuPAGE Bis-Tris SDS-PAGE gels in running buffer (50 mM MES, 50 mM Tris, 0.1% SDS, 1 mM EDTA, pH 7.3), transferred onto nitrocellulose membrane (Amersham #10600041) in transfer buffer (8 mM Tris, 30 mM glycine, 20% methanol) and stained with Ponceau S (Sigma-Aldrich, #P7170). Membranes were blocked in TBS-T+5% BSA for 1 hr at room temperature (RT) and incubated overnight at 4°C in primary antibodies prepared in TBS-T+5% BSA. Membranes were washed 3×15 min in TBS-T, incubated with secondary antibody for 1 hr at RT, washed, and imaged using Odyssey CLx (LI-COR). Antibodies: Histone H3 (Abcam, ab1791, 1:1000); Histone H4 (Abcam, ab10158, 1:1000), Vimentin (Cell Signalling Technology (CST), #5741S, 1:1000), IRDye 680RD Donkey anti-Rabbit IgG Secondary Antibody (LI-COR, 926-68073, 1:10,000), GAPDH (CST, #97166, 1:10,000), OCT4A (CST, #2840, 1:10,000), SOX2 (CST, #3579, 1:10,000), NANOG (CST, #4903, 1:10,000).

## Flow cytometric analysis

Cells were seeded on Geltrex-coated dishes in TESR medium at a density of $3\times10^3$ cells/cm$^2$. After 24 hr the medium was replaced with fresh TESR medium and after a further 24 hr the cells were harvested using TrypLE select, resuspended in TESR medium and counted (cell density at harvest was approximately $1\times10^5$ cells/cm$^2$ for each cell line). $5\times10^5$ cells were then collected by centrifugation at 300×*g* for 2 min then resuspended with 1 ml of Dulbecco's PBS (without calcium or magnesium) containing 1% fetal bovine serum. The cells were then collected by centrifugation at 300×*g* for 2 min and resuspended in 1 ml of ice-cold 90% methanol/10% dH$_2$O while vortexing. Samples were then incubated for 30 min at RT before being stored at –20°C until they were analysed.

For cell cycle analysis the cells were collected by centrifugation at 300×*g* for 2 min, washed with PBS, then resuspended in 300 µl of staining buffer (Dulbecco's PBS+1% FBS+50 µg/ml propidium iodide, 50 µg/ml ribonuclease A) for 20 min at RT in the dark. Cellular DNA content was determined by analysis on a FACSCanto Flow Cytometer (BD Biosciences). PI fluorescence was detected using 488 nm excitation and fluorescence emission collected at 585/42 nm. Data was analysed using FlowJo software. Doublet discrimination was performed on the basis of PI-A v PI-W measurements and cell cycle distribution determined using the Watson Pragmatic model.

## Cell line selection for MS

hiPSCs (bubh_3, kucg_2, oaqd_3 and wibj_2) and hESCs (SA121 and SA181, H1 and H9) were analysed by MS using TMT as described below.

## Protein extraction

Cell pellets were resuspended in 300 µl extraction buffer (4% SDS in 100 mM triethylammonium bicarbonate [TEAB], phosphatase inhibitors [PhosSTOP, Roche]). Samples were boiled (15 min, 95°C, 350 rpm) and sonicated for 30 cycles in a bath sonicator (Bioruptor Pico bath sonicator, Diagenode, Belgium; 30 s on, 30 s off) followed by probe sonication for 50 s (20 s on, 5 s off). 2 µl Benzonase nuclease HC (250 U/µl, Merck Millipore) was added and incubated for 30 min (37°C, 750 rpm). Reversibly oxidised cysteines were reduced with 10 mM TCEP (45 min, 22°C, 1000 rpm) followed by alkylation of free thiols with 20 mM iodoacetamide (45 min, 22°C, 1000 rpm, in the dark). Proteins were quantified using the fluorometric EZQ assay (Thermo Fisher Scientific).

## Protein digestion using the SP3 method

Protein extracts were cleaned and digested with the SP3 method as described previously with modifications (*Hughes et al., 2014*; *Hughes et al., 2019*). Briefly, 50 µl of a 20 µg/µl SP3 bead stock (Sera-Mag SpeedBead carboxylate-modified magnetic particles; GE Healthcare Life Sciences) and 500 µl acetonitrile (ACN; final concentration of 70%) were added to 150 µl of protein extract and incubated at RT for 10 min (1000 rpm). Tubes were mounted on a magnetic rack, supernatants were removed and beads were washed twice with 70% ethanol and once with ACN (1 ml each). Beads were resuspended in 80 µl 100 mM TEAB and digested for 4 hr with LysC followed by tryptic digestion overnight (1:50 protease:protein ratio, 37°C, 1000 rpm). Peptides were cleaned by addition of 3.5 µl formic acid (final concentration of 4%) and 1.7 ml ACN (final concentration of 95%) followed by incubation for 10 min. After spinning down (1000×*g*) tubes were mounted on a magnetic rack and beads were washed once with 1.5 ml ACN. Peptides were eluted from the beads with 100 µl 2% DMSO and acidified with 5.2 µl 20% formic acid (final concentration of 1%) followed by centrifugation (15,000×*g*). Peptide amounts were quantified using the fluorometric CBQCA assay (Thermo Scientific).

## TMT labelling

For each sample 15 µg peptides per sample were dried in vacuo in a Concentrator plus (Eppendorf) and resuspended in 50 µl 200 mM EPPS pH 8.5. TMT10plex tags (Thermo Scientific) were dissolved in anhydrous ACN and added to the peptide sample in a 1:10 peptide:TMT ratio. Additional anhydrous ACN was added to a final volume of 22 µl. Samples were incubated for 2 hr (22°C, 750 rpm). Unreacted TMT was quenched by incubation with 5 µl 5% hydroxylamine for 30 min. Samples were combined, dried in vacuo, and resuspended in 1% TFA followed by clean-up with solid-phase extraction using Waters Sep-Pak tC18 50 mg. Samples were loaded, washed five times with 1 ml 0.1% TFA in water, and peptides were eluted with 70% ACN/0.1% TFA (1 ml) and dried in vacuo in a Concentrator plus (Eppendorf).

## High pH reversed phase peptide fractionation

TMT labelled peptide samples were fractionated using off-line high pH reverse phase chromatography. Dried samples were resuspended in 5% formic acid and loaded onto a 4.6×250 mm XBridge BEH130 C18 column (3.5 µm, 130 Å; Waters). Samples were separated on a Dionex Ultimate 3000 HPLC system with a flow rate of 1 ml/min. Solvents used were water (A), ACN (B), and 100 mM ammonium formate pH 9 (C). While solvent C was kept constant at 10%, solvent B started at 5% for 3 min, increased to 21.5% in 2 min, 48.8% in 11 min and 90% in 1 min, was kept at 90% for further 5 min followed by returning to starting conditions and re-equilibration for 8 min. Peptides were separated into 48 fractions, which were concatenated into 24 fractions and subsequently dried in vacuo. Peptides were redissolved in 5% formic acid and analysed by LC-MS.

## LC-MS analysis

TMT labelled samples were analysed on an Orbitrap Fusion Tribrid mass spectrometer coupled to a Dionex RSLCnano HPLC (Thermo Scientific). Samples were loaded onto a 100 µm×2 cm Acclaim PepMap-C18 trap column (5 µm, 100 Å) with 0.1% trifluoroacetic acid for 7 min and a constant flow of 4 µl/min. Peptides were separated on a 75 µm×50 cm EASY-Spray C18 column (2 µm, 100 Å; Thermo Scientific) at 50°C using a linear gradient from 10% to 40% B in 153 min with a flow rate of 200 nl/min. Solvents used were 0.1% formic acid (A) and 80% ACN/0.1% formic acid (B). The spray was initiated by applying 2.5 kV to the EASY-Spray emitter. The ion transfer capillary temperature

was set to 275°C and the radio frequency of the S-lens to 50%. Data were acquired under the control of Xcalibur software in a data-dependent mode. The number of dependent scans was 12. The full scan was acquired in the orbitrap covering the mass range of $m/z$ 350–1400 with a mass resolution of 120,000, an AGC target of $4 \times 10^5$ ions and a maximum injection time of 50 ms. Precursor ions with charges between 2 and 7 and a minimum intensity of $5 \times 10^3$ were selected with an isolation window of $m/z$ 1.2 for fragmentation using collision-induced dissociation in the ion trap with 35% collision energy. The ion trap scan rate was set to 'rapid'. The AGC target was set to $1 \times 10^4$ ions with a maximum injection time of 50 ms and a dynamic exclusion of 60 s. During the MS3 analysis, for more accurate TMT quantification, five fragment ions were co-isolated using SPS in a window of $m/z$ 2 and further fragmented with an HCD collision energy of 65%. The fragments were then analysed in the orbitrap with a resolution of 50,000. The AGC target was set to $5 \times 10^4$ ions and the maximum injection time was 105 ms.

## TMT channel allocation

| TMT tag | Cell line |
| --- | --- |
| 126C | bubh_3 (hiPSC - female) |
| 127N | H01 (hESC - male) |
| 127C | kucg_2 (hiPSC - male) |
| 128N | H09 (hESC - female) |
| 128C | aqd_3 (hiPSC - male) |
| 129N | SA121 (hESC - male) |
| 129C | wibj_2 (hiPSC - female) |
| 130N | SA181 (hESC - male) |
| 130C | Empty |
| 131 | Empty |

## High-resolution respirometry in wibj_2 and H1 stem cells

Mitochondrial respiration was studied in digitonin-permeabilised wibj_2 and H1 stem cells (10 μg/$1 \times 10^6$ cells) with six technical replicates per cell line, to keep mitochondria in their architectural environment. The analysis was performed in an oxygraphic chamber with thermostat set to 37°C with continuous stirring (Oxygraph-2 k, Oroboros Instruments, Innsbruck, Austria). Cells were collected with trypsin, pelleted, and then placed in MiR05 respiration medium (110 mM sucrose, 60 mM lactobionic acid, 0.5 mM EGTA, 3 mM MgCl$_2$, 20 mM taurine, 10 mM KH$_2$PO$_4$, 20 mM HEPES adjusted to pH 7.1 with KOH at 30°C, and 1 g/l BSA, essentially fatty acid free). Substrate-Uncoupler-Inhibitor titration protocol number 2 (SUIT-002) (*Doerrier et al., 2018*) was used to determine respiratory rates. Briefly, after residual oxygen consumption in absence of endogenous fuel substrates, residual oxygen consumption (ROX), in presence of 2.5 mM ADP, was measured, fatty acid oxidation pathway state (F) was evaluated by adding malate (0.1 mM) and octanoyl carnitine (0.2 mM) (OctM$_P$). Membrane integrity was tested by adding cytochrome $c$ (10 μM) (OctMc$_P$). Subsequently, the NADH electron transfer pathway state (FN) was studied by adding a high concentration of malate (2 mM, OctM$_P$), pyruvate (5 mM, OctPM$_P$), and glutamate (10 mM, OctPGM$_P$). Then, succinate (10 mM, OctPGMS$_P$) was added to stimulate the S pathway (FNS), followed by glycerophosphate (10 mM, OctPGMSGp$_P$) to reach convergent electron flow in the FNSGp pathway to the Q-junction. Uncoupled respiration was next measured by performing a titration with CCCP (OctPGMSGp$_E$), followed by inhibition of complex I (SGp$_E$) with rotenone (0.5 μM, SGp$_E$). Finally, ROX was measured by adding Antimycin A (2.5 μM). ROX was then subtracted from all respiratory states, to obtain mitochondrial respiration. Results are expressed in pmol · s$^{-1}$ · $1 \times 10^6$ cells. The P/E control ratio, which reflects the control by coupling and limitation by the phosphorylation system, was subsequently calculated by dividing the OctPGMSGp$_P$ value by the OctPGMSGp$_E$ value.

## Radiolabelled glutamine uptake (protocol was adapted from *Yeramian et al., 2006*)

Two hiPSC lines (wibj_2 and oaqd_3) with three technical replicates each were compared to two hESC lines (SA121 and SA181) with three technical replicates of each. Both hiPSCs and hESCs were plated in six-well plates 2 days before the transport assay ($5×10^4$ cells/cm$^2$ – this gives $1×10^6$ cells/well on 'uptake day'). The cell growth media was carefully aspirated so as not to disturb the adherent monolayer of cells. They were washed gently by pipetting with 5 ml preheated (37°C) uptake solution (HBSS [pH 7.4], Gibco) and aspirating off. This was repeated three times. They were then incubated with 0.5 ml of uptake solution containing [$^3$H]glutamine (5 µCi/ml; Perkin Elmer, NET 55100) in either the presence, or absence, of L-glutamine (5 mM; Sigma) for 2 min.

Glutamine uptake was stopped by removing the uptake solution and washing cells with 2 ml of ice-cold stop solution (HBSS with 10 mM nonradioactive L-glutamine) three times. After the third wash, the cells were lysed in 200 µl of 0.1% SDS and 100 mM NaOH, and 100 µl was used to measure the radioactivity associated with the cells. Finally, 100 µl sample was added to scintillation vials containing 3 ml scintillant (*OptiPhase HiSafe* 3, Perkin Elmer). β-Radioactivity was measured with Tri-Carb 4910TR liquid scintillation counter.

The net glutamine CPM values were calculated by subtracting the Quench CPM values from the glutamine CPM values.

## TEM sample preparation

wibj_2 and H1 cells were fixed on the dish in 4% paraformaldehyde and 2.5% glutaraldehyde in 0.1 M sodium cacodylate buffer (pH 7.2) for 30 min then scraped and transferred to a tube and fixed for a further 30 min prior to pelleting. The pellets were cut into small pieces, washed three times in cacodylate buffer and then post-fixed in 1% OsO4 with 1.5% Na ferricyanide in cacodylate buffer for 60 min. After another three washes in cacodylate buffer they were contrasted with 1% tannic acid and 1% uranyl acetate. The cell pellets were then dehydrated through alcohol series into 100% ethanol, changed to propylene oxide left overnight in 50% propylene oxide 50% resin and finally embedded in 100% Durcupan resin (Sigma). The resin was polymerised at 60°C for 48 hr and sectioned on a Leica UCT ultramicrotome. Sections were contrasted with 3% aqueous uranyl acetate and Reynolds lead citrate before imaging on a JEOL 1200EX TEM using a SIS III camera.

## Proteomics search parameters

The data were searched and quantified with MaxQuant (*Cox and Mann, 2008*) (version 1.6.7) against the human SwissProt database from UniProt (*The UniProt, 2017*) (November 2019). The data were searched with the following parameters: type was set to Reporter ion on MS3 with 10plex TMT, stable modification of carbamidomethyl (C), variable modifications of oxidation (M), acetylation (proteins N terminus), and deamidation (NQ). The missed cleavage threshold was set to 2, and the minimum peptide length was set to 7 amino acids. The FDR was set to 1% for positive identification at the protein and peptide spectrum match level.

## Unique, shared, and razor peptides

Peptides which are exclusive to a single protein group are considered unique peptides. Peptides whose sequences match more than one protein group are called shared peptides. Razor peptides are shared peptides whose intensity gets assigned to a single protein group despite matching multiple protein groups.

## Data filtering

All protein groups identified with less than either 2 razor or unique peptides or labelled as 'Contaminant', 'Reverse', or 'Only identified by site' were removed from the analysis.

## Peptide normalisation

For *Figure 6—figure supplement 1* and *Figure 6—figure supplement 2* peptide intensities were divided by the sum of the intensity from all histone peptides and were multiplied by $1×10^6$.

## Copy number calculations

Protein copy numbers were estimated following the 'proteomic ruler' method (*Wiśniewski et al., 2014*), but adapted to work with TMT MS3 data. The summed MS1 intensities were allocated to the different experimental conditions according to their fractional MS3 reporter intensities.

## Protein content estimations

The protein content was estimated using the following formula: CN × MW and then converting the data from daltons to picograms, where CN is the protein copy number and MW is the protein molecular weight (in Da).

## Differential expression analysis

Fold changes and p-values were calculated in R. For individual proteins the p-values were calculated with the bioconductor package LIMMA (*Ritchie et al., 2015*) version 3.7. The q-values provided were generated in R using the 'qvalue' package version 2.10.0. p-Values for protein families and protein complexes were calculated in R using Welch's t-test.

## hiPSC vs hESC overrepresentation analysis

All overrepresentation analysis were done on WebGestalt. The first analysis selected proteins with a fold change>2 and a q-value<0.001. The second analysis selected proteins whose fold change was lower than the median minus one standard deviation (0.195) and a q-value<0.001. Both analyses used all identified proteins with 2 or more razor and unique peptides as a background and required an FDR lower than 0.05.

## Peptide coverage figures

*Figure 6—figure supplement 1* and *Figure 6—figure supplement 2* were generated with Protter (*Omasits et al., 2014*).

## Acknowledgements

We would like to thank Gabriel Sollberger as well as all members of the Lamond Laboratory for their input and advice. This work was supported by the Wellcome Trust/MRC grant (098503/E/12/Z), Wellcome Trust grants (073980/Z/03/Z, 105024/Z/14/Z, 206293/Z/17/Z, 097418/Z/11/Z, 205023/Z/16/Z), BBSRC Project Grant (BB/V010948/1), EPSRC grant (EP/Y010655/1), a Wellcome Trust Equipment Award (202950/Z/16/Z), and a UK Research Partnership Infrastructure Fund award to the Centre for Translational and Interdisciplinary Research.

## Additional information

### Competing interests

Eva Griesser: Now works for Boehringer Ingelheim Pharma GmbH & Co KG. Melpomeni Platani: Board member of Tartan Cell Technologies Ltd. Jason R Swedlow: Board member of Tartan Cell Technologies Ltd and Glencoe Software Ltd. Angus I Lamond: Board member of Tartan Cell Technologies Ltd and Platinum Informatics Ltd. The other authors declare that no competing interests exist.

### Funding

| Funder | Grant reference number | Author |
| --- | --- | --- |
| Wellcome Trust | 098503/E/12/Z | Angus I Lamond |
| Wellcome Trust | 10.35802/073980 | Angus I Lamond |
| Biotechnology and Biological Sciences Research Council | BB/V010948/1 | Angus I Lamond |

| Funder | Grant reference number | Author |
|---|---|---|
| Engineering and Physical Sciences Research Council | EP/Y010655/1 | Angus I Lamond |
| Wellcome Trust | 10.35802/097418 | Doreen A Cantrell |
| Wellcome Trust | 10.35802/105024 | Doreen A Cantrell Angus I Lamond |
| Wellcome Trust | 10.35802/205023 | Doreen A Cantrell |
| Wellcome Trust | 206293/Z/17/Z | Jason R Swedlow Angus I Lamond |

The funders had no role in study design, data collection and interpretation, or the decision to submit the work for publication. For the purpose of Open Access, the authors have applied a CC BY public copyright license to any Author Accepted Manuscript version arising from this submission.

## Author contributions

Alejandro J Brenes, Conceptualization, Data curation, Formal analysis, Validation, Investigation, Visualization, Writing – original draft, Project administration, Writing – review and editing, Data interpretation; Eva Griesser, Conceptualization, Methodology, Writing – review and editing; Linda V Sinclair, Validation, Methodology, Writing – review and editing, Data interpretation; Lindsay Davidson, Validation, Methodology, Writing – review and editing, Data interpretation; Alan R Prescott, Validation, Methodology, Writing – review and editing, Data interpretation; Francois Singh, Validation, Methodology, Writing – review and editing, Data interpretation; Elizabeth KJ Hogg, Validation, Methodology, Writing – review and editing, Data interpretation; Carmen Espejo-Serrano, Validation, Methodology, Writing – review and editing; Hao Jiang, Validation, Methodology, Writing – review and editing, Data interpretation; Harunori Yoshikawa, Conceptualization, Visualization, Writing – review and editing, Data interpretation; Melpomeni Platani, Methodology, Writing – review and editing, Data interpretation; Jason R Swedlow, Funding acquisition, Methodology, Writing – review and editing, Data interpretation; Greg M Findlay, Funding acquisition, Validation, Methodology, Writing – review and editing; Doreen A Cantrell, Supervision, Funding acquisition, Validation, Methodology, Writing – review and editing; Angus I Lamond, Conceptualization, Supervision, Funding acquisition, Investigation, Writing – original draft, Project administration, Writing – review and editing

## Author ORCIDs

Alejandro J Brenes ⓘ https://orcid.org/0000-0001-8298-2463
Linda V Sinclair ⓘ https://orcid.org/0000-0003-1248-7189
Alan R Prescott ⓘ https://orcid.org/0000-0002-0747-7317
Elizabeth KJ Hogg ⓘ https://orcid.org/0000-0002-2509-3202
Hao Jiang ⓘ https://orcid.org/0000-0003-4123-4930
Harunori Yoshikawa ⓘ https://orcid.org/0000-0003-3793-6219
Jason R Swedlow ⓘ https://orcid.org/0000-0002-2198-1958
Greg M Findlay ⓘ https://orcid.org/0000-0002-7222-4965
Doreen A Cantrell ⓘ https://orcid.org/0000-0001-7525-3350
Angus I Lamond ⓘ https://orcid.org/0000-0001-6204-6045

Reviewer #1 (Public review): https://doi.org/10.7554/eLife.92025.3.sa1
Reviewer #2 (Public review): https://doi.org/10.7554/eLife.92025.3.sa2
Reviewer #3 (Public review): https://doi.org/10.7554/eLife.92025.3.sa3
Author response https://doi.org/10.7554/eLife.92025.3.sa4

# Additional files

## Supplementary files
• Supplementary file 1. Protein copy numbers for all human induced pluripotent stem cells (hiPSCs) and human embryonic stem cells (hESCs).
• Supplementary file 2. Median intensity normalisation (concentration-like) based differential

expression analysis for human induced pluripotent stem cells (hiPSCs) and human embryonic stem cells (hESCs).

• Supplementary file 3. Copy number-based differential expression analysis for human induced pluripotent stem cells (hiPSCs) and human embryonic stem cells (hESCs).

• MDAR checklist

## Data availability

The proteomic raw files and the mzTab outputs were uploaded to PRIDE as a full submission under the identifier PXD014502 and are available at https://www.ebi.ac.uk/pride/archive/projects/PXD014502. The processed copy number and volcanos plot files are included in the supplemental data.

The following dataset was generated:

| Author(s) | Year | Dataset title | Dataset URL | Database and Identifier |
|---|---|---|---|---|
| Brenes AJ, Lamond AI | 2021 | TMT characterisation: iPSC vs hESC | https://www.ebi.ac.uk/pride/archive/projects/PXD014502 | PRIDE, PXD014502 |

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
