## [Editor Report · eLife Assessment]

This study reports differences in proteomic profiles of embryonic versus induced pluripotent stem cells. This **important** finding cautions against the interchangeable use of both types of cells in biomedical research, although the mechanisms responsible for these differences remains unknown. The proteomic evidence is **convincing**, even though there is limited validation with other methods.

---

## [Referee Report · Reviewer #1 (Public review)]

Summary:

The authors compared four types of hiPSCs and four types of hESCs at the proteome level to determine their differences. Semiquantitative calculations of protein copy number revealed increased protein content in iPSCs. In particular, the results suggest that mitochondria- and cytoplasm-associated proteins in iPSCs reflect to some extent the state of the original differentiated cells. Basically, it contains responses to almost all comments and adds text mainly to the discussion. No additional experiments were performed in the revision, but I believe that future validation using methods other than proteomics would provide more support for the results.

Pros:

Mitochondrial function was verified by high-resolution respirometry, indicating increased ATP-producing capacity of the phosphorylation system in iPSCs.

Weaknesses:

The proteome data in this study may be the result of a simple examination of differences between the clones, and proteome data should be verified using various methods in the future.

---

## [Referee Report · Reviewer #2 (Public review)]

Summary:

Pluripotent stem cells are powerful tools for understanding development, differentiation, and disease modeling. The capacity of stem cells to differentiate into various cell types holds great promise for therapeutic applications. However, ethical concerns restrict the use of human embryonic stem cells (hESCs). Consequently, induced human pluripotent stem cells (ihPSCs) offer an attractive alternative for modeling rare diseases, drug screening, and regenerative medicine. A comprehensive understanding of ihPSCs is crucial to establish their similarities and differences compared to hESCs. This work demonstrates systematic differences in the reprogramming of nuclear and non-nuclear proteomes in ihPSCs.

Strengths:

The authors employed quantitative mass spectrometry to compare protein expression differences between independently derived ihPSC and hESC cell lines. Qualitatively, protein expression profiles in ihPSC and hESC were found to be very similar. However, when comparing protein concentration at a cellular level, it became evident that ihPSCs express higher levels of proteins in the cytoplasm, mitochondria, and plasma membrane, while the expression of nuclear proteins is similar between ihPSCs and hESCs. A higher expression of proteins in ihPSCs was verified by an independent approach, and flow cytometry confirmed that ihPSCs had larger cell size than hESCs. The differences in protein expression were reflected in functional distinctions. For instance, the higher expression of mitochondrial metabolic enzymes, glutamine transporters, and lipid biosynthesis enzymes in ihPSCs was associated with enhanced mitochondrial potential, increased ability to uptake glutamine, and increased ability to form lipid droplets.

Weaknesses:

While this finding is intriguing and interesting, the study falls short of explaining the mechanistic reasons for the observed quantitative proteome differences. It remains unclear whether the increased expression of proteins in ihPSCs is due to enhanced transcription of the genes encoding this group of proteins or due to other reasons, for example, differences in mRNA translation efficiency. Another unresolved question pertains to how the cell type origin influences ihPSC proteomes. For instance, whether ihPSCs derived from fibroblasts, lymphocytes, and other cell types all exhibit differences in their cell size and increased expression of cytoplasmic and mitochondrial proteins. Analyzing ihPSCs derived from different cell types and by different investigators would be necessary to address these questions.

---

## [Referee Report · Reviewer #3 (Public review)]

This study provides a useful insight into the proteomic analysis of several human induced pluripotent (hiPSC) and human embryonic stem cell (hESC) lines. Although the study is largely descriptive with limited validation of the differences found in the proteomic screen, the findings provide a solid platform for further mechanistic discovery.

---

## [Author Response]

The following is the authors’ response to the original reviews.

**Public Reviews:**

**Reviewer #1 (Public Review):**
Summary:The authors compared four types of hiPSCs and four types of hESCs at the proteome level to elucidate the differences between hiPSCs and hESCs. Semi-quantitative calculations of protein copy numbers revealed increased protein content in iPSCs. Particularly in iPSCs, proteins related to mitochondrial and cytoplasmic were suggested to reflect the state of the original differentiated cells to some extent. However, the most important result of this study is the calculation of the protein copy numbers per cell, and the validity of this result is problematic. In addition, several experiments need to be improved, such as using cells of different genders (iPSC: female, ESC: male) in mitochondrial metabolism experiments.Strengths:The focus on the number of copies of proteins is exciting and appreciated if the estimated calculation result is correct and biologically reproducible.Weaknesses:The proteome results in this study were likely obtained by simply looking at differences between clones, and the proteome data need to be validated. First, there were only a few clones for comparison, and the gender and number of cells did not match between ESCs and iPSCs. Second, no data show the accuracy of the protein copy number per cell obtained by the proteome data.

We agree with the reviewer that it would be useful to have data from more independent stem cell clones and ideally an equal gender balance of the donors would be preferable. As usual, practical cost-benefit, and time available affect the scope of work that can be performed. We note that the impact of biological donor sex on proteome expression in iPSC lines has already been addressed in previous studies13. We will however revise the manuscript to include specific mention of these limitations and propose a larger-scale follow-up when resources are available.

Regarding the estimation of protein copy numbers in our study, we would like to highlight that the proteome ruler approach we have used has been employed extensively in the field previously, with direct validation of differences in copy numbers provided using orthogonal methods to MS, e.g., FACS2-4,7,10. Furthermore, the original manuscript14 directly compared the copy numbers estimated using the “proteomic ruler” to spike-in protein epitope signature tags and found remarkable concordance. This original study was performed with an older generation mass spectrometer and reduced peptide coverage, compared with the instrumentation used in our present study. Further, we noted that these authors predicted that higher peptide coverage, such as we report in our study, would further increase quantitative performance.

**Reviewer #2 (Public Review):**
Summary:Pluripotent stem cells are powerful tools for understanding development, differentiation, and disease modeling. The capacity of stem cells to differentiate into various cell types holds great promise for therapeutic applications. However, ethical concerns restrict the use of human embryonic stem cells (hESCs). Consequently, induced human pluripotent stem cells (ihPSCs) offer an attractive alternative for modeling rare diseases, drug screening, and regenerative medicine. A comprehensive understanding of ihPSCs is crucial to establish their similarities and differences compared to hESCs. This work demonstrates systematic differences in the reprogramming of nuclear and non-nuclear proteomes in ihPSCs.

We thank the reviewer for the positive assessment.

Strengths:The authors employed quantitative mass spectrometry to compare protein expression differences between independently derived ihPSC and hESC cell lines. Qualitatively, protein expression profiles in ihPSC and hESC were found to be very similar. However, when comparing protein concentration at a cellular level, it became evident that ihPSCs express higher levels of proteins in the cytoplasm, mitochondria, and plasma membrane, while the expression of nuclear proteins is similar between ihPSCs and hESCs. A higher expression of proteins in ihPSCs was verified by an independent approach, and flow cytometry confirmed that ihPSCs had larger cell sizes than hESCs. The differences in protein expression were reflected in functional distinctions. For instance, the higher expression of mitochondrial metabolic enzymes, glutamine transporters, and lipid biosynthesis enzymes in ihPSCs was associated with enhanced mitochondrial potential, increased ability to uptake glutamine, and increased ability to form lipid droplets.Weaknesses:While this finding is intriguing and interesting, the study falls short of explaining the mechanistic reasons for the observed quantitative proteome differences. It remains unclear whether the increased expression of proteins in ihPSCs is due to enhanced transcription of the genes encoding this group of proteins or due to other reasons, for example, differences in mRNA translation efficiency. Another unresolved question pertains to how the cell type origin influences ihPSC proteomes. For instance, whether ihPSCs derived from fibroblasts, lymphocytes, and other cell types all exhibit differences in their cell size and increased expression of cytoplasmic and mitochondrial proteins. Analyzing ihPSCs derived from different cell types and by different investigators would be necessary to address these questions.

We agree with the Reviewer that our study does not extend to also providing a detailed mechanistic explanation for the quantitative differences observed between the two stem cell types and did not claim to have done so. We have now included an expanded section in the discussion where we discuss potential causes. However, in our view fully understanding the reasons for this difference is likely to involve extensive future in-depth analysis in additional studies and is not something that can be determined just by one or two additional supplemental experiments.

We also agree studying hiPSCs reprogrammed from different cell types, such as blood lymphocytes, would be of great interest. Again, while we agree it is a useful way forward, in practice this will require a very substantial additional commitment of time and resources. We have now included a section discussing this opportunity within the discussion to encourage further research into the area.

**Recommendations for the authors:**

**Reviewer #1 (Recommendations For The Authors):**
(1) aizi1 and ueah1 clones, which were analyzed in Figure 1A, were excluded from the proteome analysis. In particular, the GAPDH expression level of the aizi1 clone is similar to that of ESCs and different from other iPSC clones. An explanation of how the clones were selected for proteome analysis is needed. Previously, the comparative analysis of iPSCs and ESCs reported in many studies from 2009-2017 (Ref#1-7) has already shown that the number of clones used in the comparative analysis is small, claiming differences (Ref#1-3) and that the differences become indistinguishable when the number of clones is increased (Ref#4-7). Certainly, few studies have been done at the proteome level, so it is important to examine what differences exist in the proteome. Also, it is interesting to focus on the amount of protein per cell. However, if the authors want to describe biological differences, it would be better to get the proteome data in biological duplicate and state the reason for selecting the clones used.(1) M. Chin, Cell Stem Cell, 2009, PMID: 19570518(2) K. Kim, Nat Biotechnol., 2011, PMID: 22119740(3) R. Lister, Nature, 2011, PMID: 21289626(4) A.M. Newman, Cell Stem Cell, 2010, PMID: 20682451(5) M.G. Guenther, Cell Stem Cell, 2010, PMID: 20682450(6) C. Bock, Cell, 2010, PMID: 21295703(7) S. Yamanaka, Cell Stem Cell, PMID: 22704507

We agree with the reviewer that analysing more clones would be beneficial. We have included a section of this topic in the discussion. In our study, we only had access to the 4 hESC lines included, therefore in the original proteomic study we also analysed 4 hiPSC lines, which were routinely grown within our stem cell facility. While as the study progressed the stem cell facility expanded the culture of additional hiPSC lines, unfortunately we couldn’t also access additional hESC lines.

We agree that ideally combining each biological replicate with additional technical replicates would provide extra robustness. As usual, cost and practical considerations at the time the experiments were performed affected the experimental design chosen. For the experimental design, each experiment was contained within 1 batch to avoid the strong batch effects present in TMT (Brenes et al 2019).

(2) iPSC samples used in the proteome analysis are two types of female and two types of male, while ESC samples are three types of female and one type of female. The number of sexes of the cells in the comparative analysis should be matched because sex differences may bias the results.

While we agree with the reviewer in principle, we have previously performed detailed comparisons of proteome expression in many independent iPSC lines from both biological male and female donors (see Brenes et al., Cell Reports 2021) and it seems unlikely that biological sex differences alone could account for the proteome differences between iPS and ESC lines uncovered in this study . However, as this is a relevant point, we have revised the manuscript to explicitly mention this caveat within the discussion section.

(3) In Figure 1h, I suspect that the variation of PCA plots is very similar between ESCs and iPSCs. In particular, the authors wrote "copy numbers for all 8 replicates" in the legend, but if Figure 1b was done 8 times, there should be 8 types of cells x 8 measurements = 64 points. Even if iPSCs and ESCs are grouped together, there should be 8 points for each cell type. Is it possible that there is only one TMT measurement for this analysis? If so, at least technical duplicates or biological duplicates would be necessary. I also think each cell should be plotted in the PCA analysis instead of combining the four types of ESCs and iPSCs into one.

We thank the reviewer for bringing this error to our attention. The legend has been corrected to state, “for all 8 stem cell lines”. Each dot represents the proteome of each of the 4 hESCs and 4 hiPSCs that were analysed using proteomics.

(4) It is necessary to show what functions are enriched in the 4408 proteins whose protein copies per cell were increased in the iPSCs obtained in Figure 2B.

The enrichment analysis requested has been performed and is now included as a new supplemental figure 2. We find it very interesting that despite the large number of proteins involved here (4,408), the enrichment analysis still shows clear enrichment for specific cellular processes. The summary plot using affinity propagation within webgestalt is included here:

**Author response image 1. sa4fig1:** 

(5) The Proteomic Ruler method used in this study is a semi-quantitative method to calculate protein copy numbers and is a concentration estimation method. Therefore, if the authors want to have a biological discussion based on the results, they need to show that the estimated concentrations are correct. For example, there are Western Blotting (WB) results for genes with no change in protein levels in hESC and hiPSC in Fig. 6ij, but the WB results for the group of genes that are claimed to have changed are not shown throughout the paper. Also, there is no difference in the total protein level between iPSCs and ESCs from the ponceau staining in Fig.6ij. WB results for at least a few genes are needed to show whether the concentration estimates obtained from the proteome analysis are plausible. If the protein per cell is increased in these iPSC clones, performing WB analysis using an equal number of cells would be better.

Regarding the ‘proteome ruler’ approach we would like to highlight that this method has previously been used extensively in the field, with detailed validation, as already explained above. It is also not ‘semi-quantitative’ and can estimate absolute abundance, as well as concentrations. Our work does not use their concentration formulas, but the estimation of protein copy numbers, which was shown to closely match the observed copy numbers as determined when spike-ins are used14.

In providing here additional validation using Western Blotting (WB), we prioritised for analysis also by WB the proteins related to pluripotency markers, which are vital to determine the pluripotency state of the hESCs and hiPSCs, as well as histone markers. We have included a section in the discussion concerning additional validation data and agree in general that further validation is always useful.

(6) Regarding the experiment shown in Figure 4l, the gender of iPSC used (wibj2) is female and WA01 (H1; WA01) is male. Certainly, there is a difference in the P/E control ratio, but isn't this just a gender difference? The sexes of the cells need to be matched.

We accept that ideally the sexes of donors should ideally have been matched and have mentioned this within the discussion. Nonetheless, as previously mentioned, our previous detailed proteomic analyses of multiple hiPSC lines13 derived from both biological male and female donors provide relevant evidence that the results shown in this study are not simply a reflection of the sex of the donors for the respective iPSC and ESC lines. When comparing eroded and non-eroded female hiPSCs to male hiPSCs we found no significant differences in any electron transport chain proteins, not TCA proteins between males and females.

Minor comments:(1) Method: Information on the hiPSCs and hESCs used in this study should be described. In particular, the type of differentiated cells, gender, and protocols that were used in the reprogramming are needed.

We agree with the reviewer on this. The hiPSC lines were generated by the HipSci consortium, as described in the flagship HipSci paper15. We cite the flagship paper, which specifies in great detail the reprogramming protocols and quality control measures, including analysis of copy number variations15. However, we agree that this information may not be easily accessible for readers. We agree it is relevant to explicitly include this information in our present manuscript, instead of expecting readers to look at the flagship paper. These details have therefore been added to the revised version.

(2) Method: In Figure1a, Figure 6i, j, the antibody information of Nanog, Oct4, Sox2, and Gapdh is not written in the method and needs to be shown.

The data relating to these has now been included within the methods section.

(3) Method: In Figure 1b and other figures, the authors should indicate which iPSC corresponds to which TMT label; the data in the Supplemental Table also needs to indicate which data is which clone.

We have now added this to the methods section.

(4) Method: The method of the FACS experiment used in Figure 2 should be described.

The methods related to the FACS analysis have now been included within the manuscript.

(5) Method: The cell name used in the mitochondria experiment shown in Figure 4 is listed as WA01, which is thought to be H1. Variations in notation should be corrected.

This has now been corrected.

(6) Method: The name of the cell clone shown in Figure 3l,m should be mentioned.

We have now added these details on the corresponding figure and legend.

**Reviewer #2 (Recommendations For The Authors):**
This study utilized quantitative mass spectrometry to compare protein expression in independently derived 4 ihPSC and 4 hESC cell lines. The investigation quantified approximately 7,900 proteins, and employing the "Proteome ruler" approach, estimated protein copy numbers per cell. Principal component analyses, based on protein copy number per cell, clearly separated hiPSC and hESC, while different hiPSCs and hESCs grouped together. The study revealed a global increase in the expression of cytoplasmic, mitochondrial, membrane transporters, and secreted proteins in hiPSCs compared to hESCs. Interestingly, standard median-based normalization approaches failed to capture these differences, and the disparities became apparent only when protein copy numbers were adjusted for cell numbers. Increased protein abundance in hiPSC was associated with augmented ribosome biogenesis. Total protein content was >50% higher in hiPSCs compared to hESCs, a observation independently verified by total protein content measurement via the EZQ assay and further supported by the larger cell size of hiPSCs in flow cytometry. However, the cell cycle distribution of hiPSC and hESC was similar, indicating that the difference in protein content was not due to variations in the cell cycle. At the phenotypic level, differences in protein expression also correlated with increased glutamine uptake, enhanced mitochondrial potential, and lipid droplet formation in hiPSCs. ihPSCs also expressed higher levels of extracellular matrix components and growth factors.Overall, the presented conclusions are adequately supported by the data. Although the mechanistic basis of proteome differences in ihPSC and hESC is not investigated, the work presents interesting findings that are worthy of publication. Below, I have listed my specific questions and comments for the authors.(1) Figure 1a displays immunoblots from 6 iPSC and 4 ESC cell lines, with 8 cell lines (4 hESC, 4 hiPSC) utilized in proteomic analyses (Fig. 1b). The figure legend should specify the 8 cell lines included in the proteomic analyses. The manuscript text describing these results should explicitly mention the number and names of cell lines used in these assays.

We agree with the reviewer and have now marked in figure 1 all the lines that were used for proteomics and have added a section in the methods specifying which cell lines were analysed in each TMT channel.

(2) In most figures, the quantitative differences in protein expression between hiPSC and hESC are evident, and protein expression is highly consistent among different hiPSCs and hESCs. However, the glutamine uptake capacity of different hiPSC cell lines, and to some extent hESC cell lines, appears highly variable (Figure 3e). While proteome changes were measured in 4 hiPSCs and 4 hESCs, the glutamine uptake assays were performed on a larger number of cell lines. The authors should clarify the number of cell lines used in the glutamine uptake assay, clearly indicating the cell lines used in the proteome measurements. Given the large variation in glutamine uptake among different cell lines, it would be useful to plot the correlation between the expression of glutamine transporters and glutamine uptake in individual cell lines. This may help understand whether differences in glutamine uptake are related to variations in the expression of glutamine transporters.

The “proteomic ruler” has the capacity to estimate the protein copy numbers per cell, as such changes in the absolute number of cells that were analysed do not cause major complications in quantification. Furthermore, TMT-based proteomics is the most precise proteomics methods available, where the same peptides are detected in all samples across the same data points and peaks, as long as the analysis is done within a single batch, as is the case here.

The glutamine uptake assay is much more sensitive to the variation in the number of cells. The number of cells were estimated by plating the cells with approximately 5e4 cells two days before the assay, which creates variability. Furthermore, hESCs and hiPSCs are more adhesive than the cells used in the original protocol, hence the quench data was noisier for these lines, making the data from the assay more variable.

(3) In Figure 4j, it would be helpful to indicate whether the observed differences in the respiration parameters are statistically significant.

We have now modified the plot to show which proteins were significantly different.

(4) The iPSCs used here are generated from human primary skin fibroblasts. Different cells vary in size; for instance, fibroblast cells are generally larger than blood lymphocytes. This raises the question of whether the parent cell origin impacts differences in hiPSCs and hESC proteomes. For example, do the authors anticipate that hiPSCs derived from small somatic cells would also display higher expression of cytoplasmic, mitochondrial, and membrane transporters compared to ESC? The authors may consider discussing this point.

This is a very interesting point. We have now added an extension to the discussion focussed on this subject.

(5) One wonders if the "Proteome ruler" approach could be applied retrospectively to previously published ihPSC and hESC proteome data, confirming higher expression of cytoplasmic and mitochondrial proteins in ihPSCs, which may have been masked in previous analyses due to median-based normalization.

We agree with the reviewer and think this is a very good suggestion. Unfortunately, in the main proteomic papers comparing hESC and hiPSCs16,17 the authors did not upload their raw files to a public repository (as it was not mandatory at that period in time), and they also used the International Protein Index (IPI), which is a discontinued database. So the raw files can’t be reprocessed and the database doesn’t match the modern SwissProt entries. Therefore, reprocessing the previous data was impractical.

(6) The work raises a fundamental question: what is the mechanistic basis for the higher expression of cytoplasmic and mitochondrial proteins in ihPSCs? Conceivably, this could be due to two reasons: (a) Genes encoding cytoplasmic and mitochondrial proteins are expressed at a higher level in ihPSCs compared to hESC. (b) mRNAs encoding cytoplasmic and mitochondrial proteins are translated at a higher level in ihPSCs compared to hESC. The authors may check published transcriptome data from the same cell lines to shed light on this point.

This is a very interesting point. We believe that the reprogrammed cells contained mature mitochondria, which are not fully regressed upon reprogramming and that this can establish a growth advantage in the normoxic environments in which the cells are grown. Unfortunately, the available transcriptomic data lacked spike-ins, and thus only enables comparison of concentration, not of copy numbers13. Therefore, we could not determine with the available data if there was an increase in the copies of specific mRNAs. However, with a future study where there was a transcriptomic dataset with spike-ins included, this would be very interesting to analyse.

**Reviewer #3 (Recommendations For The Authors):**
It is unclear whether changes in protein levels relate to any phenotypic features of cell lines used. For example, the authors highlight that increased protein expression in hiPSC lines is consistent with the requirement to sustain high growth rates, but there is no data to demonstrate whether hiPSC lines used indeed have higher growth rates.

We respectfully disagree with the reviewer on this point. Our data show that hESCs and hiPSCs show significant differences in protein mass and cell size, with the MS data validated by the EZQ assay and FACS, while having no significant differences in their cell cycle profiles. Thus, increased size and protein content would require higher growth rates to sustain the increased mass, which is what we observe.

The authors claim that the cell cycle of the lines is unchanged. However, no details of the method for assessing the cell cycle were included so it is difficult to appreciate if this assessment was appropriately carried out and controlled for.

We apologise for this omission; the details have been included in the revised version of the manuscript.

Details and characterisation of iPSC and ESC lines used in this study are overall lacking. The lines used are merely listed in methods, but no references are included for published lines, how lines were obtained, what passage they were used at, their karyotype status etc. For details of basic characterisation, the authors should refer to the ISSC Standards for the use of human stem cells in research. In particular, the authors should consider whether any of the changes they see may be attributed to copy number variants in different lines.

We agree with the reviewer on this and refer to the reply above concerning this issue.

The expression data for markers of undifferentiated state in Figure 1a would ideally be shown by immunocytochemistry or flow cytometry as it is impossible to tell whether cultures are heterogeneous for marker expression.

We agree with the reviewer on this. FACS is indeed much more quantitative and a better method to study heterogeneity. However, we did not have protocols to study these markers using FACS.

TEM analysis should ideally be quantified.

We agree with the reviewer that it would be nice to have a quantitative measure.

All figure legends should explicitly state what graphs are representing (e.g. average/mean; how many replicates (biological or technical), which lines)? Some data is included in Methods (e.g. glutamine uptake), but not for all of the data (e.g. TEM).

We agree with the reviewer. These has been corrected in the revised version of the manuscript, with additional details included.

Validation experiments were performed typically on one or two cell lines, but the lines used were not consistent (e.g. wibj_2 versus H1 for respirometry and wibj_2, oaqd_3 versus SA121 and SA181 for glutamine uptake). Can the authors explain how the lines were chosen?

The validation experiments were performed at different time points, and the selection of lines reflected the availability of hiPSC and hESC lines within our stem cell facility at a given point in time.

We chose to use a range of different lines for comparison, rather than always comparing only one set of lines, to try to avoid a possible bias in our conclusions and thus to make the results more general.

The authors should acknowledge the need for further functional validation of the results related to immunosuppressive proteins.

We agree with the reviewer and have added a sentence in the discussion making this point explicitly.

Differences in H1 histones abundance were highlighted. Can the authors speculate as to the meaning of these differences?

Regarding H1 histones, our study of the literature, as well as discussions with with chromatin and histone experts, both within our institute and externally, have not shed light into what the differences could imply, based upon previous literature. We think therefore that this is a striking and interesting result that merits further study, but we have not yet been able to formulate a clear hypothesis on the consequences.

(1) Howden, A. J. M. *et al.* Quantitative analysis of T cell proteomes and environmental sensors during T cell differentiation. *Nat Immunol*, doi:10.1038/s41590-019-0495-x (2019).

(2) Marchingo, J. M., Sinclair, L. V., Howden, A. J. & Cantrell, D. A. Quantitative analysis of how Myc controls T cell proteomes and metabolic pathways during T cell activation. *Elife*
**9**, doi:10.7554/eLife.53725 (2020).

(3) Damasio, M. P. *et al.* Extracellular signal-regulated kinase (ERK) pathway control of CD8+ T cell differentiation. *Biochem J*
**478**, 79-98, doi:10.1042/BCJ20200661 (2021).

(4) Salerno, F. *et al.* An integrated proteome and transcriptome of B cell maturation defines poised activation states of transitional and mature B cells. *Nat Commun*
**14**, 5116, doi:10.1038/s41467-023-40621-2 (2023).

(5) Antico, O., Nirujogi, R. S. & Muqit, M. M. K. Whole proteome copy number dataset in primary mouse cortical neurons. *Data Brief*
**49**, 109336, doi:10.1016/j.dib.2023.109336 (2023).

(6) Edwards, W. *et al.* Quantitative proteomic profiling identifies global protein network dynamics in murine embryonic heart development. *Dev Cell*
**58**, 1087-1105 e1084, doi:10.1016/j.devcel.2023.04.011 (2023).

(7) Barton, P. R. *et al.* Super-killer CTLs are generated by single gene deletion of Bach2. *Eur J Immunol*
**52**, 1776-1788, doi:10.1002/eji.202249797 (2022).

(8) Phair, I. R., Sumoreeah, M. C., Scott, N., Spinelli, L. & Arthur, J. S. C. IL-33 induces granzyme C expression in murine mast cells via an MSK1/2-CREB-dependent pathway. *Biosci Rep*
**42**, doi:10.1042/BSR20221165 (2022).

(9) Niu, L. *et al.* Dynamic human liver proteome atlas reveals functional insights into disease pathways. *Mol Syst Biol*
**18**, e10947, doi:10.15252/msb.202210947 (2022).

(10) Murugesan, G., Davidson, L., Jannetti, L., Crocker, P. R. & Weigle, B. Quantitative Proteomics of Polarised Macrophages Derived from Induced Pluripotent Stem Cells. *Biomedicines*
**10**, doi:10.3390/biomedicines10020239 (2022).

(11) Ryan, D. G. *et al.* Nrf2 activation reprograms macrophage intermediary metabolism and suppresses the type I interferon response. *iScience*
**25**, 103827, doi:10.1016/j.isci.2022.103827 (2022).

(12) Nicolas, P. *et al.* Systems-level conservation of the proximal TCR signaling network of mice and humans. *J Exp Med*
**219**, doi:10.1084/jem.20211295 (2022).

(13) Brenes, A. J. *et al.* Erosion of human X chromosome inactivation causes major remodeling of the iPSC proteome. *Cell Rep*
**35**, 109032, doi:10.1016/j.celrep.2021.109032 (2021).

(14) Wisniewski, J. R., Hein, M. Y., Cox, J. & Mann, M. A "proteomic ruler" for protein copy number and concentration estimation without spike-in standards. *Mol Cell Proteomics*
**13**, 3497-3506, doi:10.1074/mcp.M113.037309 (2014).

(15) Kilpinen, H. *et al.* Common genetic variation drives molecular heterogeneity in human iPSCs. *Nature*
**546**, 370-375, doi:10.1038/nature22403 (2017).

(16) Phanstiel, D. H. *et al.* Proteomic and phosphoproteomic comparison of human ES and iPS cells. *Nat Methods*
**8**, 821-827, doi:10.1038/nmeth.1699 (2011).

(17) Munoz, J. *et al.* The quantitative proteomes of human-induced pluripotent stem cells and embryonic stem cells. *Mol Syst Biol*
**7**, 550, doi:10.1038/msb.2011.84 (2011).